# A Comparative Analysis of Weighting Methods in Geospatial Flood Risk Assessment: A Trinidad Case Study

Cassie Roopnarine [1] , Bheshem Ramlal [1] and Ronald Roopnarine [2],*

1   Department of Geomatics and Land Management, Faculty of Engineering, The University of the West Indies, St. Augustine 685509, Trinidad and Tobago

2   Department of Food Production, Faculty of Food and Agriculture, The University of the West Indies, St. Augustine 685509, Trinidad and Tobago

*   Correspondence: ronald.roopnarine@sta.uwi.edu

**Abstract:** The Republic of Trinidad and Tobago is an archipelagic Small Island Developing State (SIDS), situated on the southern end of the chain of Caribbean islands. Several factors such as climate, topography, and hydrological characteristics increase its susceptibility and vulnerability to flooding which results in adverse socio-economic impacts. Many Caribbean islands, including Trinidad and Tobago lack a flood risk assessment tool which is essential for a proactive mitigation approach to floods, specifically in the Caribbean due to the incommensurate flooding events that occur because of the inherent characteristics of SIDS. This research focuses on the problem of flooding using susceptibility analysis, vulnerability analysis and risk assessment for the island of Trinidad, whilst also presenting a repeatable and appropriate methodology to assess these risks in regions that have similar characteristics to Trinidad. This is especially useful in Caribbean countries because of a lack of internal human capacity to support such efforts. Flood hazard indexes (FHI) and vulnerability indexes (VI) were generated for this study using subjective and objective weighting technique models to identify regions that are affected by flooding. These models were Analytical Hierarchy Process (AHP), Frequency Ratio (FR) and Shannon's Entropy (SE). Comparative analyses of the three models were conducted to assess the efficacy and accuracy of each to determine which is most suitable. These were used to conduct a risk assessment to identify risks associated with each Regional Corporation of Trinidad. Results indicate that FR is the most accurate weighting technique model to assess flood susceptibility and risk assessment in Trinidad, with an Area Under the Curve (AUC) of 0.76 and 0.64 respectively. This study provides an understanding of the most appropriate weighting techniques that can be used in regions where there are challenges in accessing comprehensive data sets and limitations as it relates to access to advanced technology and technical expertise. The results also provide reasonably accurate outcomes that can assist in identifying priority areas where further quantitative assessments may be required and where mitigation and management efforts should be focused. This is critical for SIDS where vulnerability to flooding is high while access to financial and human resources is limited.

**Keywords:** flood risk assessment; disaster risk resilience; natural hazards; Caribbean SIDS; analytical hierarchy process; frequency ratio; Shannon's entropy; GIS

## 1. Introduction

Small island developing states (SIDS) have recognizable differences in terms of geo-physical and socio-economic characteristics however, they face similar risks associated with the impacts of climate change. In most cases, they have high exposure to natural hazards, specifically water-related hazards combined with very vulnerable societies [1]. Narrow resource bases, the dominance of economic sectors that are reliant on the natural environment, limited industrial activity, physical remoteness and limited economies of scale are inherent to SIDS, thus leading to the recognition of these nations as a special

group [2]. The close relationship that exists in SIDS between people and coastal environments increases the exposure to risks associated with rapid and slow onset events such as sea level rise and flooding [3]. It is expected that climate change will further impact physical parameters, precipitation and extreme storms, increasing their frequency, thereby leading to more projected flooding events [2].

The vulnerability of Caribbean Small Island Developing States (SIDS) to the impacts of climate change and resulting hydro-climatic extremes has been well documented in recent years [4,5]. The steep topography, intense tropical rainfall, poor land and soil management, and limited enforcement of environmental policies and laws are key factors that render SIDS particularly vulnerable to natural hazards. Flooding in the Caribbean is predicted to intensify exponentially due to unregulated urbanization of floodplains, catchment degradation because of anthropogenic activity, lack of preparedness and resilience for emergency response, poverty, lack of implementation of public policies and infrastructural problems [6]. Unfortunately, most Caribbean SIDS also do not have the internal human capacity to undertake much of the work needed to support mitigation efforts [7].

Flooding is one of the major hazards that affect the islands annually, as a result of an increased occurrence of intense rainfall events and storm surges from hurricanes [8]. Trinidad is one of the islands that has been significantly impacted by flooding events of varying intensities and frequencies. Flooding is a recurring issue which occurs predominantly during the rainy season due to an increase in rainfall events [9,10]. Anthropogenic influences such as the replacement of permeable surfaces with installations of human-made channels, deforestation, and hillside development have all contributed to increased overland flow, ultimately increasing the intensity and frequency of flooding events. Of 7000 natural disasters recorded, 75 percent were related to water events, of which floods were the most frequent [8]. Several studies have highlighted the need for more emphasis to be placed on disaster risk management in the Caribbean [11] that focuses on data-driven risk interventions. In most cases, investigations have been superficial and not at a scale suitable to inform community-level actions.

Because of the destruction and damage created by floods, that disrupt the livelihoods and the economies of a country, flood risk management at a community level is an essential tool. It allows for the minimization of the destruction and the more effective management of flood risks. In the context of this study, the risk is defined as "The probability of harmful consequences or expected losses resulting from a given hazard to a given element at danger or peril, over a specified time period" [12]. Flood risk assessments involve the sequential examination of flood susceptibility, which is used interchangeably with flood hazards that can be delivered through Flood Hazard Indexes (FHIs). Flood susceptibility is defined as a "quantitative or qualitative assessment of classification, area and spatial distribution of flood" [12].

Another integral element of flood risk is flood vulnerability, which focuses on examining the factors that render a community, system or asset susceptible to adverse impacts of flooding events. Vulnerability is the conditions determined by physical, social, economic and environmental factors or processes which increase the susceptibility of an individual, a community, assets or systems to the impacts of hazards [13]. Vulnerability to a flood is multi-faceted and should incorporate physical, environmental, social, and economic factors which allow for easy quantification of vulnerability classes [14]. Research by The World Bank [10] suggested that accounting for flood hazards and the impacts that they can create, is crucial for flood hazard and risk assessment. Currently, there are limited numbers of flood risk assessments conducted in the Caribbean.

A study was conducted by Pinos and Quesada-Roman [6] to investigate flood risk-related research trends in Latin America and the Caribbean (LAC) (2000–2020). The study identified hydrometeorological assessments, flood risk analyses, flood vulnerability and resilience approaches, and statistical and geographic information sciences as the main flood studies conducted in the Caribbean. Highlighted, were several journals that contain no records of flood-related articles in the past 2–5 years. Whilst the study illustrated that most

articles of the 302 investigated were related to flood hazards and risk assessment, most of the risk studies were recorded for Mexico, Brazil, Chile, Peru and Argentina. Additionally, identified was the disproportionate production of articles that exist, with the Caribbean placing last in the LAC for the number of risk assessment studies conducted.

Several approaches have been used for risk assessments, flood susceptibility mapping and vulnerability assessments in the Caribbean. [15] developed a generic multi-hazard risk assessment model that could be applicable to the Caribbean using LATIS method, redesigned for an area in Jamaica. Whilst this method generated vulnerability maps, flood hazard maps could not be generated, and risk assessment could not be computed because of a lack of data to support the LATIS method. Jetten [16] used LISEM open-source integrated watershed erosion/runoff model. This model was developed in the Netherlands and utilized discharge, elevation, river systems, soils, land use and infrastructure as inputs into the model. The downfall of using this technique is its applicability to Saint Lucia, the lack of discharge data. Rainfall data was utilized to simulate the flash flooding process, and the lack of factors that are necessary for modelling floods. This method is difficult to reproduce in SIDS in the context of its diverse and unique characteristics as well as its data scarcity. Research has been conducted in Trinidad and Tobago to assess flood susceptibility and risk mapping for Trinidad using inundation factors [17]. The method used past flood occurrences obtained from the ODPM that did not consider the dimensions or intensities of the flood. This study uses frequency distribution, while being easier to conduct, is not the most accurate method. In addition, the census data used was a 2000 dataset which is outdated and is not a true reflection of the current population size. While the study sought to address the issue of flooding, it utilizes one model justified in its appropriateness to the region. Similar to this, is a study conducted by [18], which uses AHP to analyze flooding in the San Juan/Laventille region. This method utilized one model for its analysis and conducted the study in a localized region of Trinidad, diminishing its usefulness and relevance to other SIDS. Generally, studies in the region focused on using a single model and in some cases were limited to a small geographic region. Modeling approaches were also unimodal and with no attempt to assess relative accuracies. These methods have limitations as it relates to the characteristics of SIDS, including Trinidad.

On a global scale numerous advanced, data-intensive technology-driven techniques have been used for flood assessment studies, however many of these cannot be effectively applied to Caribbean SIDS. In the region hydraulic and hydrological modelling is limited as the data pertaining to hydrological components such as total rainfall depth, duration, peak rainfall intensity and average rainfall rate are not readily available and in instances where it is, coverage is minimal [19]. Although machine learning (ML) methods can be applied for flood susceptibility analysis using techniques such as random forest, artificial neural networks, maximum entropy or support vector machines [20,21], some challenges exist. While ML may benefit and increase the knowledge of flood simulations in a more automated way, that allows for greater profitability and efficiency in flood forecasting [22], the major issue with utilizing ML is the lack of data. ML algorithms require data to learn and predict events, due to the lack of data and expansive amounts required to increase the reliability of the process, challenges arise when utilizing these methods in developing countries that have data scarcity.

Over the past decade, GIS has been introduced as an important risk assessment tool used in spatial decision support systems. The application of GIS techniques allows for the identification, quantification, and evaluation of risk through geographical representation. By representing various spatial information, patterns and trends become apparent. Representation is done by overlaying different datasets to examine patterns and causes of spatial phenomena. Risk assessments have been conducted in the Caribbean in the past decade as the importance has been noted by multiple global investigations of flooding. Several multi-hazard risk assessments have been conducted within the Caribbean that uses GIS-based flood risk tools. Multiple weighting techniques have been utilized in the

Caribbean for hazard assessments which include logistic regression, principal component analysis, frequency distribution, weighted overlay and AHP [15,17,18].

It has been noted that challenges arise surrounding the selection of appropriate criteria weighting methods and their complexity [14] (Rincón, Khan and Armenakis 2018). Whilst Multi-Criteria Decision Making (MCDM) methods can be used to address these limitations, some MCDM can exclude social considerations which is necessary for risk assessments. AHP is considered one of the most common methods used and can be easily facilitated with GIS and pairwise comparison for hierarchal representation of the problem [23,24]. AHP is a flexible and simple tool that allows for easier manipulation. For these reasons, it was selected as the MCDM method used for risk assessment. FR has been identified as a useful tool that can determine the relationship between dependent and independent variables [23,25,26]. This bivariate study allows for past occurrences to predict future events and is best suited on a regional level. This method was selected because of the suitability of the analysis for the size of the region under investigation. SE, similar to FR, effectively identifies variables that are more influential in event occurrences and is subjective, which makes it suitable for determining flood susceptibility in the Trinidad context.

There has been research conducted related to risk assessment or risk mapping in the Caribbean. However, these works do not incorporate all the necessary steps [26] of a risk assessment to comprehensively analyze data and assess flooding in the region. As such, this research utilized a comprehensive framework for identifying disasters or hazards and assessing floods in flood-prone areas by comparing several weighing techniques. The aim of this study is to compare the output FHI and VI for Trinidad acquired from the utilization of the following weighting techniques, AHP, FR and SE which were selected based on past research conducted for areas with similar characteristics to Trinidad. The results from both FHI and final Risk Assessments were validated using the area under the curve (AUC) to assess the accuracy of each model using past flood occurrences. The study highlights which weighting technique is best suited for risk assessments, based on the selected conditioning and vulnerability factors. The latter will allow stakeholders and governmental bodies not only in Trinidad, but other Caribbean SIDS, to implement contextually relevant policies and mitigation strategies. The use of GIS tools and techniques in this study will clearly identify areas of risk that require prioritization.

Careful consideration of factors that influence a particular region needs to be identified before conducting these assessments. Several factors have been used in past research to assess flood susceptibility such as rainfall, an important climatic factor that influences the percolation and by extension the runoff in an area [14]. Topographic factors such as slope and elevation are also impactful. Higher slope values are representative of steeper regions which impact surface runoff, percolation and the velocity at which water is carried down channel networks. Elevation impacts flooding since there is a greater chance of flooding in lower-lying areas due to gravity influencing the flow of water from high to lower regions. Pallard, Castellarin, and Montanari [27], states that drainage density is crucial for flood susceptibility since it increases in drainage basins where there are great amounts of branched drainage networks and also flood peaks are related to the number of drainage networks. Voljtek and Voljtekova [20] identified the importance of using distance from rivers as a conditioning factor for flooding. They state that it highlights the surrounding regions that are closest to the rivers that are the main pathways to flooding. Most studies have used a maximum distance limit of 2 km. Flow accumulation is an important factor to be considered when conducting flood susceptibility analysis [28]. Voljtek and Voljtekova [20] state that the raster used to represent flow accumulation displays a discharge profile of the entire region for each cell. Additionally, Riadi et al. [29] describe TWI as an index that indicates the potential of an area to flood. It is an output from the amalgamation of DEM data, flow accumulation and slope. Since the factor incorporates a slope, it allows for the indication of accumulation of water at certain points as it relates to gravity. Various land uses have different impacts on the occurrence of floods. Forested areas are known for their regulation of stream flow, a decrease in run-off and overall low

susceptibility to flooding [30]. Panahi, Alijani, and Moham-madi [31] has highlighted the impacts of the shift of land use from forested areas to agriculture. Soil moisture retention capacity is impacted due to agricultural practices, resulting in an increase in surface runoff and an increase in floods. Riadi et al. [29] discuss the effect of improper anthropogenic practices conducted to remove vegetative areas for urban growth that have resulted in an increase in flood occurrences. Panahi, Alijani, and Mohammadi [31] discuss CN, stating that it is an indication of the combined effect of soil characteristics and land use cover. It is an indication of the surface runoff potential of a region.

Lithology focuses on the physical characteristic of rocks in a particular region. Various geological rocks are known to have various permeability, outcrop sizes, the thickness of layers and infiltration rates [20]. Poorer permeability results in greater runoff potentials and thereby results in greater susceptibility to flooding [32].

Factors of the study area are required to represent the social and economic vulnerabilities of Trinidad. The selection of the factors was based on the spatial scale of the study area as well as the intended purpose of the analysis and the data available. These factors are used alongside multiple flood vulnerability weighting techniques in a GIS environment to highlight vulnerable areas on a national scale in Trinidad. The conduct of vulnerability assessment in the literature focuses on similar and few factors. The factors used for this study were adapted from a study conducted by Rincón, Khan and Armenakis [14]. The social vulnerability factor used is population density. The economic vulnerability factors used are population density, building density, and road density. It was assumed that higher population densities are linked to regions with lower-income settlements that are unplanned and placed closer to drainage networks. The drainage networks in this region can also be limited, and the overcrowding of the settlements increases the drainage discharge [14]. Individuals in these areas are more likely to be impacted by flood occurrences and are at higher risk. Buildings and roads can be negatively impacted by flood occurrences. Damage to these structures implicates negatively the economic losses that ensued from flooding [14].

Flood Risk Assessments in Caribbean SIDS require localized input factors and evaluation at a micro-scale. Assessment of weighting techniques is also crucial as relatively minute differences in accuracy can translate into considerable differences at the community level in these islands. This study examines the relative accuracies of flood risk assessment outcomes using common weighting techniques at the community level in a geophysically unique region. While on a global scale there are more refined and technological approaches, this is important for SIDS given the unique circumstances and vulnerability to climate change and consequent hazards.

## 2. Study Area

The area under investigation for this study is the island of Trinidad (Figure 1). Although Trinidad has a tropical climate and is a small island (approximately 5000 square kilometers), there are still significant variations in the annual average temperatures and precipitation amounts. The Northern Range of Trinidad has a temperature ranging between 15–32 °C. Temperature and rainfall are important elements that influence Central Trinidad. Temperature varies in their mean monthly maximum throughout the year, with a mean annual temperature of 26 °C in the lowlands and lower in the highlands. High levels of rainfall of about 90–120 inches (2286–3048 mm) a year occur in the eastern region of Central Trinidad, which is referred to as the wet belt. The climate in the south of Trinidad has the driest climate where the average annual rainfall is below 80 inches (2032 mm), except for the narrow strip on the east coast and the northern flanks of the Southern Range. The mean annual temperature in South Trinidad is 26 °C. The geology of Trinidad is different from the other Caribbean islands. There are three mountain ranges in Trinidad that exhibit various types of rocks and soils that are determined by four main natural factors, relief and physiography, geology, climate, and vegetation. Figure 2 shows the distribution of the population using a choropleth map. It is estimated that the average population growth rate of Trinidad is 0.32% (World Bank 2018). Between 2000 and 2020, the population increased

from 1.27 million to 1.37 million [33]. This increase in population is synchronous with the increase in development throughout the country.

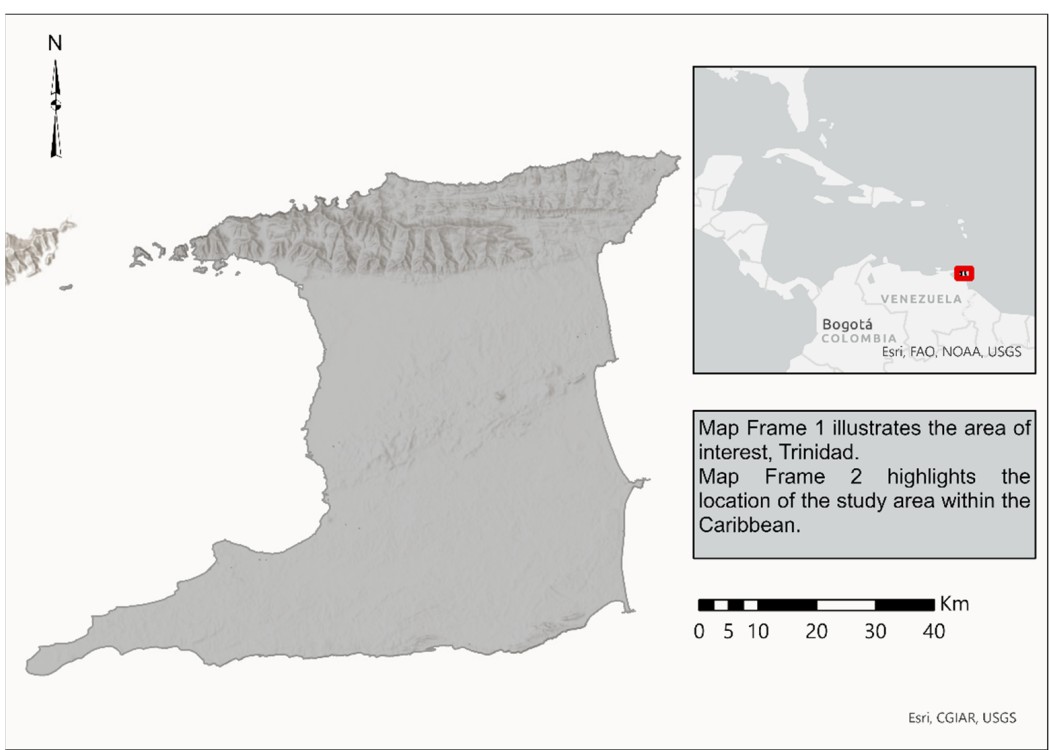

**Figure 1.** Map showing study area used.

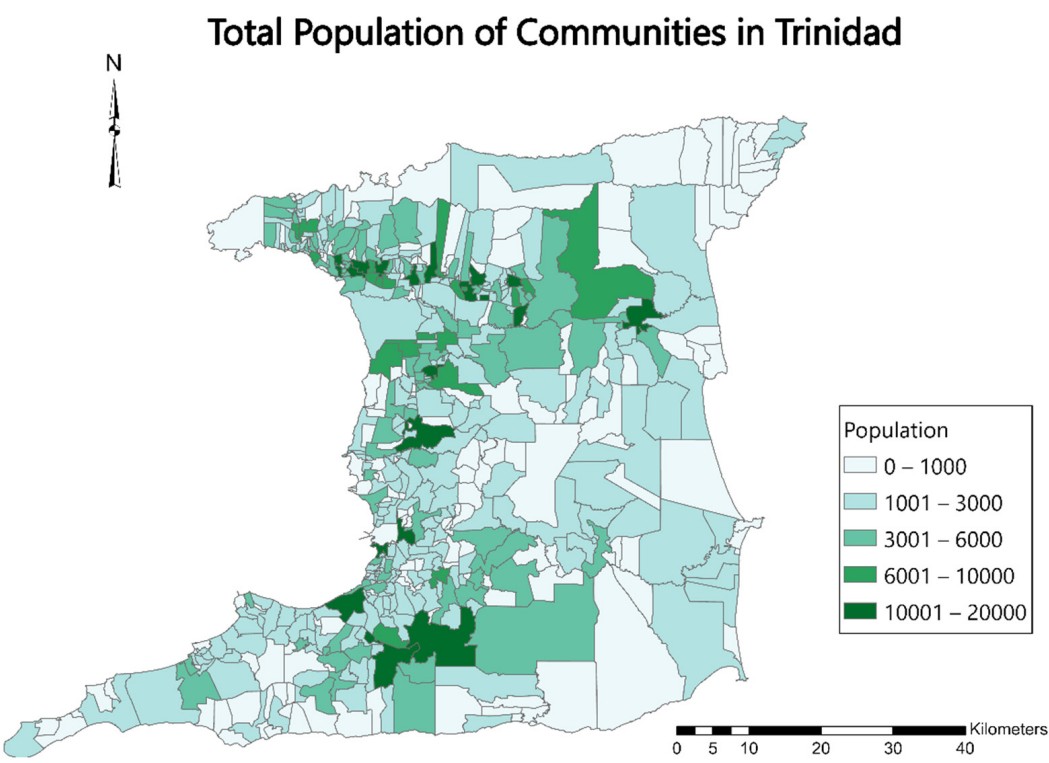

**Figure 2.** Population distribution in Trinidad.

## 3. Methodology

The analysis involves identifying what are the probable hazards or problems; determining the magnitude of the hazards; identifying the consequences and or elements at risk; the probability of damage associated with both hazard and vulnerability; and the significance of the estimated risk. An inductive approach to assess risk was taken using the following conceptual framework (Figure 3) to identify the regions at risk in Trinidad.

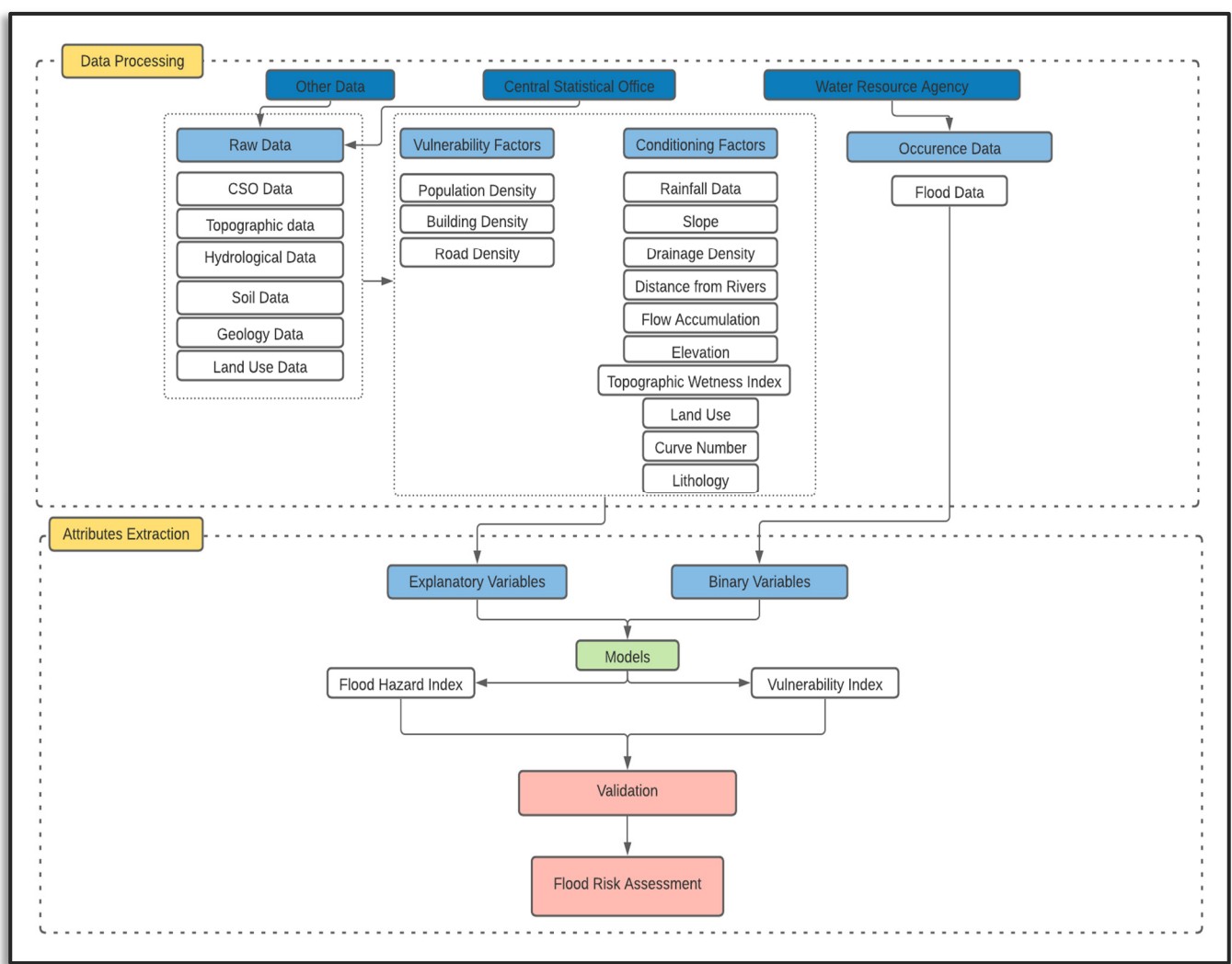

**Figure 3.** Conceptual framework for Trinidad Risk Assessment.

### 3.1. Data Sources and Risk Assessment Process

Primary and secondary data were used for this analysis (Table 1). An integrated dataset was created using digital information from the Office of Disaster Preparedness and Management (ODPM), the Central Statistical Office (CSO), The University of the West Indies (UWI), and National Aeronautics and Space Administration (NASA). These data sets included rainfall, flood occurrences, rivers, elevation, land use, geology, soils, coastline of Trinidad, buildings footprints, population, roads, community boundaries and watershed boundaries. Pre-processing and processing were conducted in ArcGIS (Esri) software. The UTM Zone 20 N/WGS84 reference system and a spatial resolution of 30 m were used for all datasets.

**Table 1.** Datasets used and its metadata.

| Data Theme | Scale of Map | Year of Production | Areal Covered | Source Agency | Data Source |
|---|---|---|---|---|---|
| Rainfall | 20 Year Average | 2020 | Trinidad | NASA | TRMM |
| Flood occurrences | 1:25,000 | 2000–2019 | Trinidad | ODPM | Field data collection |
| Rivers | 1:25,000 | 2000 | Trinidad | UWI | Digitized for existing topographic maps series |
| Elevation | 1:25,000 | 2000 | Trinidad | UWI | Digitized for existing topographic maps series |
| Land use | 1:25,000 | 2007 | Trinidad | UWI | Digitized from 1 m IKONOS Ortho-Imagery |
| Geology | 1:50,000 | 2000 | Trinidad | UWI | Digitized from 1959 Geological Map Series |
| Soils | 1:25,000 | 2000 | Trinidad | UWI | Digitized from 1975 Soil map series |
| Coastline | 1:25,000 | 1994 | Trinidad | UWI | Digitized from 1994 1 m orthophotomosaic |
| Building | 1:25,000 | 2007 | Trinidad | UWI | Digitized from 1 m IKONOS Ortho-Imagery |
| Population | 1:25,000 | 2011 | Trinidad | CSO | Compiled from the 2011 Population and Household Census |
| Roads | 1:25,000 | 2007 | Trinidad | UWI | Digitized from 1 m IKONOS Ortho-Imagery |
| Communities | 1:25,000 | 2011 | Trinidad | CSO | Compiled from the 2011 Population and Household Census |
| Watersheds | 1:25,000 | 2000 | Trinidad | UWI | Generated from 2000 Elevation dataset |

The method used for risk assessment was conducted through data processing and attribute extraction. The first stage involved pre-processing using the datasets to create layers (Figures 4–6) for conditioning factors (rainfall, slope, river density, distance from rivers, flow accumulation, elevation, topographic wetness index, land use, curve number and lithology) and vulnerability factors (population density, building density and road density). These factors were then used to assess the hazard and vulnerabilities present in Trinidad. This is conducted using three weighting techniques: AHP, FR and SE.

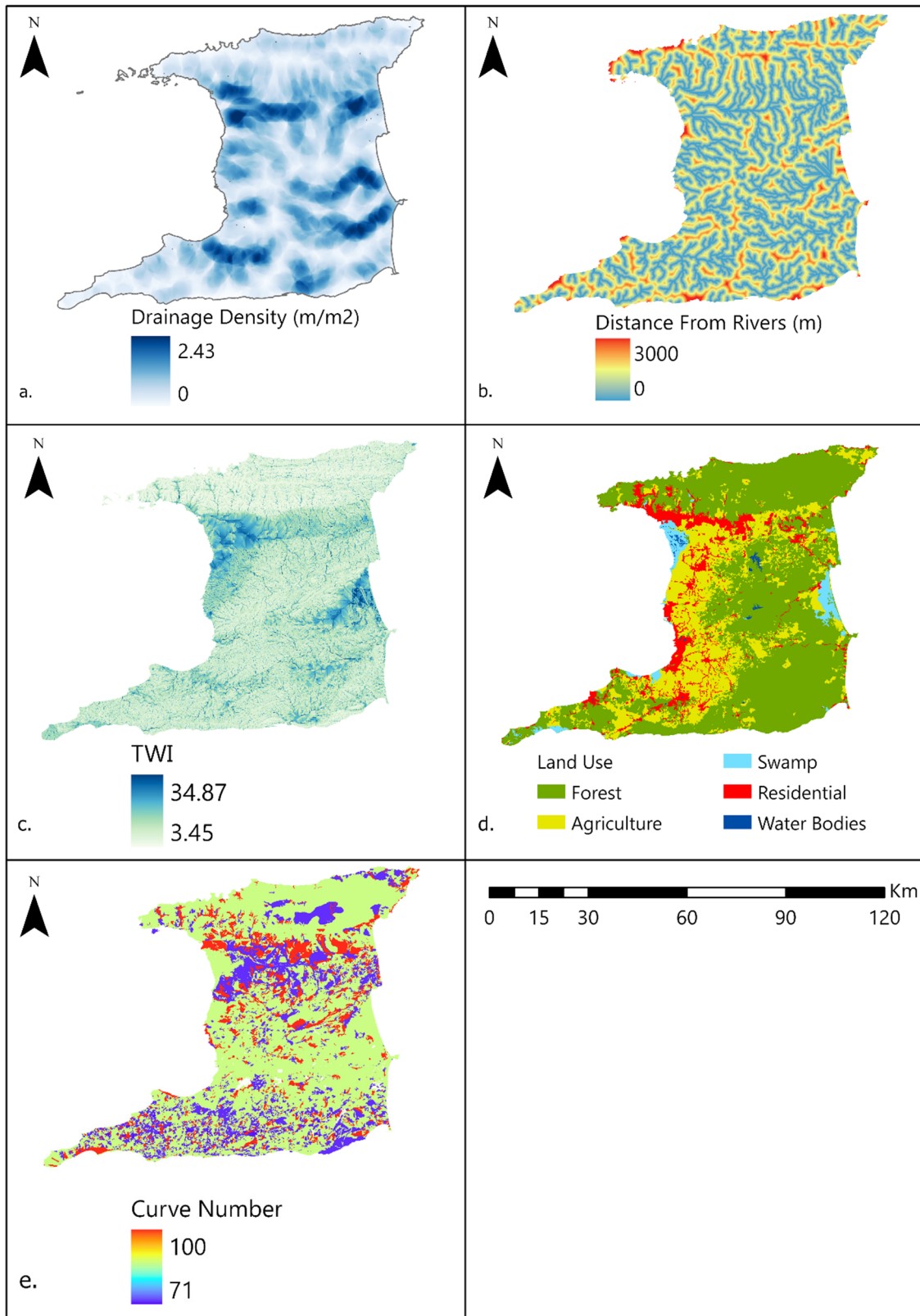

**Figure 4.** Conditioning factors used (**a**–**e**): (**a**) drainage density; (**b**) distance from rivers; (**c**) topographic wetness index; (**d**) land use; (**e**) curve number.

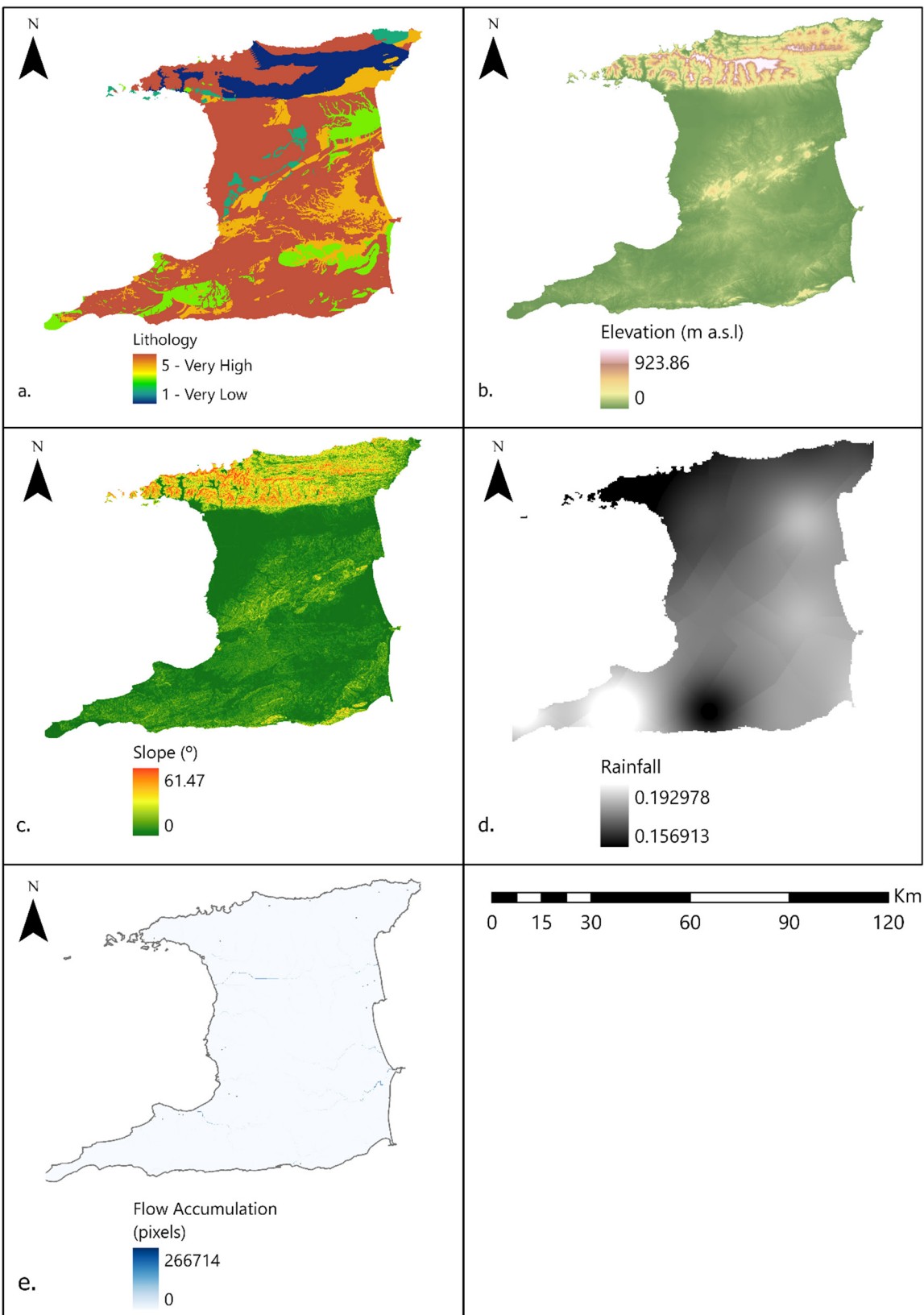

**Figure 5.** Conditioning factors used (**a–e**): (**a**) lithology; (**b**) elevation; (**c**) slope; (**d**) rainfall; (**e**) flow accumulation.

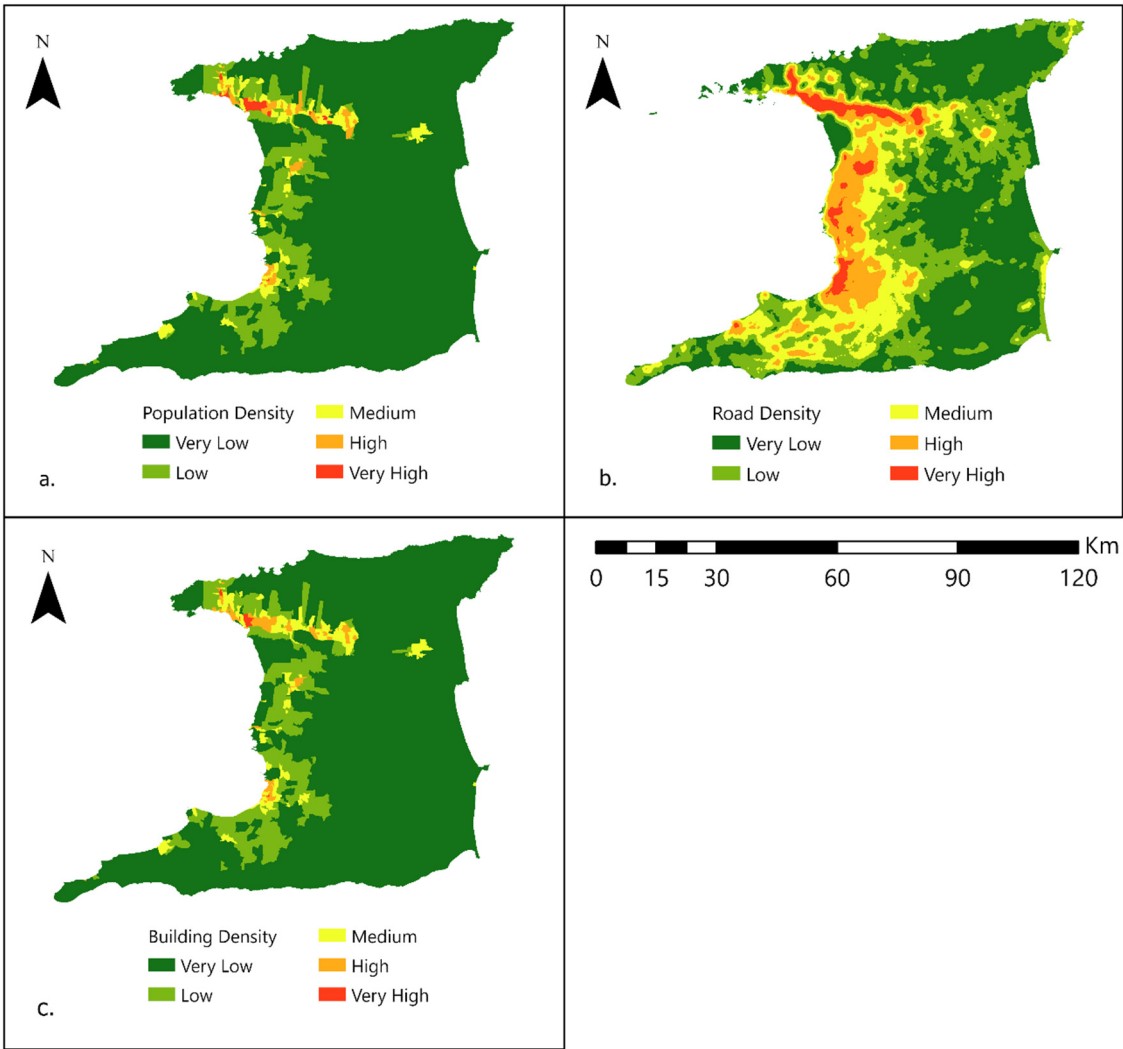

**Figure 6.** Vulnerability factors used (**a**–**c**): (**a**) Population density; (**b**) Road density; (**c**) Building density.

### 3.2. Classification and Ranking

The conditioning and vulnerability factors were reclassified into five classes to allow for comparison using Jenks natural break which optimizes the arrangement of the values into natural and optimal classes [17]. Rainfall data (mm/h) was reclassified with increasing values signifying higher ranking; slope was reclassified by degrees, with increasing slope signifying lower ranking; river density was reclassified by m/m$^2$, with increasing river density signifying higher ranking; distance from rivers (m) was reclassified with increasing distance signifying lower ranking; flow accumulation derived from DEM and reclassified by pixel values, with increasing pixel values signifying higher ranking; elevation was reclassifies using meters, with increasing height above sea level signifying lower ranking; TWI was reclassified using its raster values, with increasing values signifying higher ranking; land use was reclassified based on established norms in existing literature, with more urbanized areas ranking higher; curve number was reclassified based on the type of hydrological soil group, with soils that had more runoff potential receiving a higher ranking; lithology was reclassified using geology type, with increasing permeability signifying a lower ranking. Similarly, vulnerability factors were reclassified using their density values, with higher density signifying a higher ranking. Classification of the various layers was conducted using a classification scheme of 1 to 5 which corresponds to "very low", "low", "moderate", "high" and "very high" (Table 2). These conditioning factors were subjected to

three weighting techniques, Analytical Hierarchy Process (AHP), Frequency Ratio (FR) and Shannon's Entropy (SE).

**Table 2.** Classes of conditioning factors and rankings.

| Factors | Class | Rankings |
|---|---|---|
| **Rainfall** | 0.156913–0.176599 | 1 |
| | 0.176599–0.180521 | 2 |
| | 0.180521–0.183573 | 3 |
| | 0.183573–0.187452 | 4 |
| | 0.187452–0.192978 | 5 |
| **Slope** | 0–3.133817 | 5 |
| | 3.133817–8.919326 | 4 |
| | 8.919326–16.8744 | 3 |
| | 16.8744–26.275851 | 2 |
| | 26.275851–61.471027 | 1 |
| **River Density** | 0–0.000155 | 1 |
| | 0.000155–0.000388 | 2 |
| | 0.000388–0.000628 | 3 |
| | 0.000628–0.001002 | 4 |
| | 0.001002–0.0018 | 5 |
| **Distance from Rivers** | 0–663.59069 | 1 |
| | 663.59069–1600.424606 | 2 |
| | 1600.424606–2576.293267 | 3 |
| | 2576.293267–3649.748796 | 4 |
| | 3649.748796–4976.930176 | 5 |
| **Flow Accumulation** | 0–8367.498039 | 1 |
| | 8367.498039–37653.741176 | 2 |
| | 37653.741176–104593.72549 | 3 |
| | 104593.72549–180947.145098 | 4 |
| | 180947.145098–266714 | 5 |
| **Elevation** | 0–57.967819 | 5 |
| | 57.967819–152.165526 | 4 |
| | 152.165526–293.462086 | 3 |
| | 293.462086–481.857499 | 2 |
| | 481.857499–923.862122 | 1 |
| **Topographic Wetness Index** | 3.445429–7.018834 | 1 |
| | 7.018834–9.113589 | 2 |
| | 9.113589–11.701227 | 3 |
| | 11.701227–15.02819 | 4 |
| | 15.02819–34.866749 | 5 |

**Table 2.** *Cont.*

| Factors | Class | Rankings |
|---|---|---|
| | Water Bodies | 5 |
| | Swamp | 4 |
| **Land Use** | Urban | 3 |
| | Agriculture | 2 |
| | Forest | 1 |
| | Other Values | 1 |
| | 71 | 2 |
| **Curve Number** | 72 | 3 |
| | 87 | 4 |
| | 100 | 5 |
| | Excellent | 1 |
| | Good | 2 |
| **Lithology** | Fair | 3 |
| | Marginal | 4 |
| | Poor | 5 |

*3.3. Weighting Techniques*

All weights obtained from the weighting techniques were applied to the conditioning factors and vulnerability factors prepared using a weighted overlay or raster calculator in ArcMap to prepare Flood Hazard and Vulnerability Indexes which supported the conduct of a flood susceptibility and vulnerability assessment. The outputs from these assessments were then used to prepare risk assessments.

3.3.1. Analytical Hierarchy Process

Based on existing published research and empirical knowledge, a weighted linear combination approach was utilized in this study. Firstly, the problems and objectives are defined; the relative importance value of each criterion or factor based on their relation to each other is determined; pairwise comparison is conducted to produce comparison matrices; use the eigenvalue technique to determine the weights of the factors; consistency index is computed; lastly an overall rating for the factors by combining the weighted decision factors is obtained. The relative importance assigned allows for the ranking of each factor which influences the weights given (Tables 3 and 4).

**Table 3.** Normalized pairwise matrix highlighting criteria weights used for conditioning factors.

| Factors | Rainfall | Slope | River Density | Distance from Rivers | Flow Accumulation | Elevation | TWI | Land Use | CN | Lithology | Criteria Weights |
|---|---|---|---|---|---|---|---|---|---|---|---|
| **Rainfall** | 0.34 | 0.41 | 0.39 | 0.35 | 0.30 | 0.27 | 0.24 | 0.22 | 0.20 | 0.18 | 0.29 |
| **Slope** | 0.17 | 0.34 | 0.26 | 0.26 | 0.24 | 0.22 | 0.21 | 0.19 | 0.18 | 0.16 | 0.22 |
| **River Density** | 0.11 | 0.17 | 0.13 | 0.17 | 0.18 | 0.18 | 0.17 | 0.16 | 0.15 | 0.15 | 0.16 |
| **Distance from Rivers** | 0.09 | 0.11 | 0.06 | 0.09 | 0.12 | 0.13 | 0.14 | 0.14 | 0.13 | 0.13 | 0.11 |
| **Flow Accumulation** | 0.07 | 0.09 | 0.04 | 0.04 | 0.06 | 0.09 | 0.10 | 0.11 | 0.11 | 0.11 | 0.08 |
| **Elevation** | 0.05 | 0.07 | 0.03 | 0.03 | 0.03 | 0.04 | 0.07 | 0.08 | 0.09 | 0.09 | 0.06 |
| **TWI** | 0.05 | 0.05 | 0.03 | 0.02 | 0.02 | 0.02 | 0.03 | 0.05 | 0.07 | 0.07 | 0.04 |
| **Land Use** | 0.04 | 0.05 | 0.02 | 0.02 | 0.02 | 0.01 | 0.02 | 0.03 | 0.04 | 0.05 | 0.03 |
| **CN** | 0.04 | 0.04 | 0.02 | 0.01 | 0.01 | 0.01 | 0.01 | 0.01 | 0.02 | 0.04 | 0.02 |
| **Lithology** | 0.03 | 0.04 | 0.02 | 0.01 | 0.01 | 0.01 | 0.01 | 0.01 | 0.01 | 0.02 | 0.02 |

**Table 4.** Normalized pairwise matrix highlighting criteria weights used for vulnerability factors.

| NORMALIZED PAIRWISE MATRIX | | | | |
|---|---|---|---|---|
| **Factors** | **Population Density** | **Building Density** | **Road Density** | **Criteria Weights** |
| **Population Density** | 0.546 | 0.571 | 0.500 | 0.539 |
| **Building Density** | 0.273 | 0.286 | 0.333 | 0.297 |
| **Road Density** | 0.180 | 0.143 | 0.167 | 0.163 |

AHP can generate inconsistencies during the pairwise comparison, therefore, to ensure there is consistency throughout the weights, the consistency ratio was calculated (Tables 5 and 6) using Equations (1) and (2).

$$CR = \frac{CI}{RI} \tag{1}$$

where,

$CR \rightarrow$ Consistency Ratio, which is the consistency of judgements across all pairwise comparisons

$RI \rightarrow$ Random Index, determined by the number of factors

$$CI = \frac{\lambda - n}{n - 1} \tag{2}$$

where,

$CI \rightarrow$ Consistency Index
$\lambda \rightarrow$ Arithmetic mean of the consistency vector
$n \rightarrow$ Total number of factors

The *RI* was selected based on the number of factors. 1.49 and 0.58 were used as the *RI* for conditioning factors and vulnerability factors respectively. The weighted overlay tool was used to aggregate the various factors to create a flood hazard index and vulnerability index.

**Table 5.** Variables used for consistency ratio of susceptibility analysis.

| RI | 1.49 |
|---|---|
| n | 10.00 |
| lmax | 10.56 |
| CI | 0.06 |
| CR | 0.04 |

**Table 6.** Variables used for consistency ratio of vulnerability analysis.

| RI | 0.58 |
|---|---|
| n | 3 |
| lmax | 3.006 |
| CI | 0.002 |
| CR | 0.004 |

3.3.2. Frequency Ratio

This method is a bi-variate statistical model that used past occurrences in conjunction with a subjective method to map susceptibility. It assumes that the past will determine what is to be expected in the future. The flood occurrence dataset obtained from the ODPM was split into two subsets. 70 % of the total flood points were used for training data and

30% was used for testing data. Training data was utilized to conduct the FR method, whilst the test data was used to validate the results obtained.

Equation (3) was used to ascertain the *FR*, *RF* and *PR* values adapted from Tehrany et al. [34].

$$FR = \frac{(N_{pix}(SX_i)/\sum_{i=1}^{m} SX_i)}{(N_{pix}(X_j)/\sum_{j=1}^{n} X_j)} \tag{3}$$

where,

$N_{pix}(SX_i) \rightarrow$ Number of flooded pixels in the class *i* of factor *X*

$N_{pix}(X_j) \rightarrow$ Total number of pixels within each factor $X_j$

$m \rightarrow$ Number of classes in each factor $X_i$

$n \rightarrow$ Number of factors used in the study

The flood hazard index was calculated using Equation (4):

$$FHI = \sum_{j=1}^{n} FR \tag{4}$$

All conditioning factors were then reclassified using the integer of the RF (relative frequency) values calculated using Equation (5).

$$RF = \frac{factor\ class\ FR}{\sum\ factor\ class} \tag{5}$$

To create the FHI and VI for the FR method, all factors were combined in ArcMap and assigned the predictor weights of the various factors. This step is used to create susceptibility and vulnerability layers. The predictor weights can be seen in Tables 7 and 8 and was calculated using Equation (6).

$$PR = \frac{Max\ RF - Min\ RF}{Max - Min\ Min\ RF} \tag{6}$$

**Table 7.** Frequency ratios and predictive weights for the factors used in FR model for susceptibility analysis.

| | Slope | | | Topographic Wetness Index | |
|---|---|---|---|---|---|
| **Class** | **FR** | **PR** | **Class** | **FR** | **PR** |
| 1 | 0.06 | | 1 | 0.24 | |
| 2 | 0.12 | | 2 | 0.72 | |
| 3 | 0.24 | 5.72 | 3 | 1.84 | 2.85 |
| 4 | 0.42 | | 4 | 2.04 | |
| 5 | 1.64 | | 5 | 2.6 | |
| | **River Density** | | | **Land Use** | |
| **Class** | **FR** | **PR** | **Class** | **FR** | **PR** |
| 1 | 0.87 | | 1 | 0.06 | |
| 2 | 0.89 | | 2 | 0.74 | |
| 3 | 1.07 | 1 | 3 | 0.12 | 8.11 |
| 4 | 1.45 | | 4 | 8.29 | |
| 5 | 0.99 | | 5 | 0 | |
| | **Distance from River** | | | **Curve Number** | |
| **Class** | **FR** | **PR** | **Class** | **FR** | **PR** |
| 1 | 0.13 | | 1 | 0.29 | |
| 2 | 0.36 | | 2 | 0.62 | |
| 3 | 0.68 | 3.57 | 3 | 1.44 | 2.49 |
| 4 | 1.02 | | 4 | 0.79 | |
| 5 | 1.65 | | 5 | 1.61 | |

**Table 7.** *Cont.*

| Flow Accumulation | | | Lithology | | |
|---|---|---|---|---|---|
| **Class** | **FR** | **PR** | **Class** | **FR** | **PR** |
| 1 | 0.99 | | 1 | 0.5 | |
| 2 | 1.91 | | 2 | 1.16 | |
| 3 | 5.44 | 3.98 | 3 | 0.93 | 1.38 |
| 4 | 3.99 | | 4 | 0.62 | |
| 5 | 0 | | 5 | 1.16 | |
| **Elevation** | | | **Rainfall** | | |
| **Class** | **FR** | **PR** | **Class** | **FR** | **PR** |
| 1 | 0 | | 1 | 3.36 | |
| 2 | 0.05 | | 2 | 17.79 | |
| 3 | 0.03 | 7.15 | 3 | 1.94 | 6.61 |
| 4 | 0.28 | | 4 | 1.09 | |
| 5 | 1.38 | | 5 | 0.03 | |

**Table 8.** Frequency ratios and predictive weights for the factors used in FR model used for vulnerability analysis.

| Factors | Factor Classes | FR |
|---|---|---|
| Road Density | 1 | 0.23 |
| | 2 | 1.00 |
| | 3 | 2.27 |
| | 4 | 3.00 |
| | 5 | 0.19 |
| Building Density | 1 | 0.28 |
| | 2 | 3.42 |
| | 3 | 6.13 |
| | 4 | 11.12 |
| | 5 | 11.07 |
| Population Density | 1 | 0.28 |
| | 2 | 3.55 |
| | 3 | 6.23 |
| | 4 | 9.51 |
| | 5 | 10.39 |

### 3.3.3. Shannon's Entropy

The method identifies variables that are more influential in event occurrences and is a subjective method. The values obtained from the *FR* technique were used for the input for this technique. Equations (7)–(11) were used for the method.

$$E_{ij} = \frac{FR}{\sum_{j=1}^{M_j} FR} \tag{7}$$

where,

$FR \rightarrow$ represents the frequency ratio

$E_{ij} \rightarrow$ is the probability density for each class.

$$H_j = \sum_{i=1}^{M_j} E_{ij} log_2 E_{ij}, \ j = 1, \ldots, n \tag{8}$$

$$H_{jmax} = log_2 M_j, \ M_{j-number\ of\ classes} \tag{9}$$

$$I_j = \left( H_{max} - \frac{H_j}{H_{jmax}} \right), \ I = (0,1), \ j = 1, \tag{10}$$

$$V_j = I_j j E ij \tag{11}$$

where,

　　$H_j$ and $H_{max}$ The values of entropy

　　$I_j \rightarrow$ Information coefficient,

　　$M_j \rightarrow$ Number of classes in each conditioning factor

　　$V_j \rightarrow$ Achieved weight value for the given parameter

　　NB. The range of variation is between zero (0) and one (1). The closer the values are to one, signifies more inconsistency and imbalance.

　　To create the FHI and VI for the SE method, all factors were combined in ArcMap and assigned the predictor weights of the various factors (Tables 9 and 10). This step is used to create susceptibility and vulnerability layers.

**Table 9.** Weights given to conditioning factors of SE model.

| | Slope | | | | Topographic Wetness Index | | |
|---|---|---|---|---|---|---|---|
| **Class** | **Pij** | **Ej** | **Wj %** | **Class** | **Pij** | **Ej** | **Wj %** |
| 1 | 0.03 | −0.04 | | 1 | 0.03 | 0.00 | |
| 2 | 0.05 | −0.06 | | 2 | 0.10 | −0.10 | |
| 3 | 0.10 | −0.10 | 0.1106 | 3 | 0.25 | −0.15 | 0.0865 |
| 4 | 0.17 | −0.13 | | 4 | 0.27 | −0.15 | |
| 5 | 0.66 | −0.12 | | 5 | 0.35 | −0.16 | |
| | **River Density** | | | | **Land Use** | | |
| **Class** | **Pij** | **Ej** | **Wj %** | **Class** | **Pij** | **Ej** | **Wj %** |
| 1 | 0.16 | −0.13 | | 1 | 0.01 | 0.00 | |
| 2 | 0.17 | −0.13 | | 2 | 0.08 | −0.09 | |
| 3 | 0.20 | −0.14 | 0.0581 | 3 | 0.01 | −0.02 | 0.1765 |
| 4 | 0.28 | −0.15 | | 4 | 0.90 | −0.04 | |
| 5 | 0.19 | −0.14 | | 5 | 0.00 | 0.00 | |
| | **Distance from River** | | | | **Curve Number** | | |
| **Class** | **Pij** | **Ej** | **Wj %** | **Class** | **Pij** | **Ej** | **Wj %** |
| 1 | 0.03 | 0.00 | | 1 | 0.06 | −0.07 | |
| 2 | 0.09 | 0.00 | | 2 | 0.13 | −0.12 | |
| 3 | 0.18 | 0.00 | 0.1757 | 3 | 0.30 | −0.16 | 0.0701 |
| 4 | 0.27 | 0.00 | | 4 | 0.17 | −0.13 | |
| 5 | 0.43 | −0.16 | | 5 | 0.34 | −0.16 | |
| | **Flow Accumulation** | | | | **Lithology** | | |
| **Class** | **Pij** | **Ej** | **Wj %** | **Class** | **Pij** | **Ej** | **Wj %** |
| 1 | 0.08 | 0.00 | | 1 | 0.11 | −0.11 | |
| 2 | 0.16 | −0.13 | | 2 | 0.27 | −0.15 | |
| 3 | 0.44 | −0.16 | 0.1132 | 3 | 0.21 | −0.14 | 0.0612 |
| 4 | 0.32 | −0.16 | | 4 | 0.14 | −0.12 | |
| 5 | 0.00 | 0.00 | | 5 | 0.27 | −0.15 | |
| | **Elevation** | | | | **Rainfall** | | |
| **Class** | **Pij** | **Ej** | **Wj %** | **Class** | **Pij** | **Ej** | **Wj %** |
| 1 | 0.00 | 0.00 | | 1 | 0.14 | −0.12 | |
| 2 | 0.03 | −0.04 | | 2 | 0.73 | −0.10 | |
| 3 | 0.02 | −0.03 | 0.1479 | 3 | 0.08 | −0.09 | 0.129 |
| 4 | 0.16 | −0.13 | | 4 | 0.04 | −0.06 | |
| 5 | 0.79 | −0.08 | | 5 | 0.00 | 0.00 | |

**Table 10.** Weights given to vulnerability factors of SE model.

| Road Density | | | |
|---|---|---|---|
| **Class** | **Pij** | **Ej** | **Wj %** |
| 1 | 0.03 | −0.05 | |
| 2 | 0.15 | −0.12 | |
| 3 | 0.34 | −0.16 | 0.26 |
| 4 | 0.45 | −0.16 | |
| 5 | 0.03 | −0.04 | |
| **Building Density** | | | |
| **Class** | **Pij** | **Ej** | **Wj %** |
| 1 | 0.01 | −0.02 | |
| 2 | 0.11 | −0.10 | |
| 3 | 0.19 | −0.14 | 0.24 |
| 4 | 0.35 | −0.16 | |
| 5 | 0.35 | −0.16 | |
| **Population Density** | | | |
| **Class** | **Pij** | **Ej** | **Wj %** |
| 1 | 0.01 | 0.00 | |
| 2 | 0.12 | 0.00 | |
| 3 | 0.21 | 0.00 | 0.50 |
| 4 | 0.32 | 0.00 | |
| 5 | 0.35 | −0.16 | |

### 3.3.4. Validation

The Area under the Receiver Operating Curve (AUC–ROC) was used as an evaluation metric to determine the performance of binary classifiers and the success rates of the various models. Similar approaches were used by [26,35–38]. This step was conducted using the sensitivity value (true positive rate) plotted on the y-axis and using the value 1—Specify value (positive negative rate) on the x-axis. The main steps involved in calculating the success rate were calculating the index values (area under the curve) and sorting them in descending order; dividing the ordered cells into 100 classes with accumulated 1% intervals; recalculating to represent the total area as 1 which means perfect prediction accuracy, if the index value equals 0.5, it represents lack of fitness of the model.

## 4. Results and Discussion

### 4.1. Flood Susceptibility

After combining all the factors, three final FHIs (Figures 7–9) were generated for the analysis using AHP, FR and SE. These layers were used to compute the final risk assessments. The FHIs were created using the ten factors that were found to be major influences of flooding in Trinidad at the national level. FHI depicted five susceptibility classes, namely, very low, low, moderate, high and very high susceptibility to flooding.

### 4.1.1. FHI-AHP

From Figure 7, it is observed that over half (53.84%) of the area of Trinidad is classified as having high and very high susceptibility to flooding. The very low susceptible regions are found in areas with high elevations, such as the various ranges in Trinidad. The output displays the great influence of slope and drainage density. These two factors were given weights of 0.29 and 0.22 (Table 3) which were the two highest weights assigned to all the factors and as such would exert the most influence on the final susceptibility map.

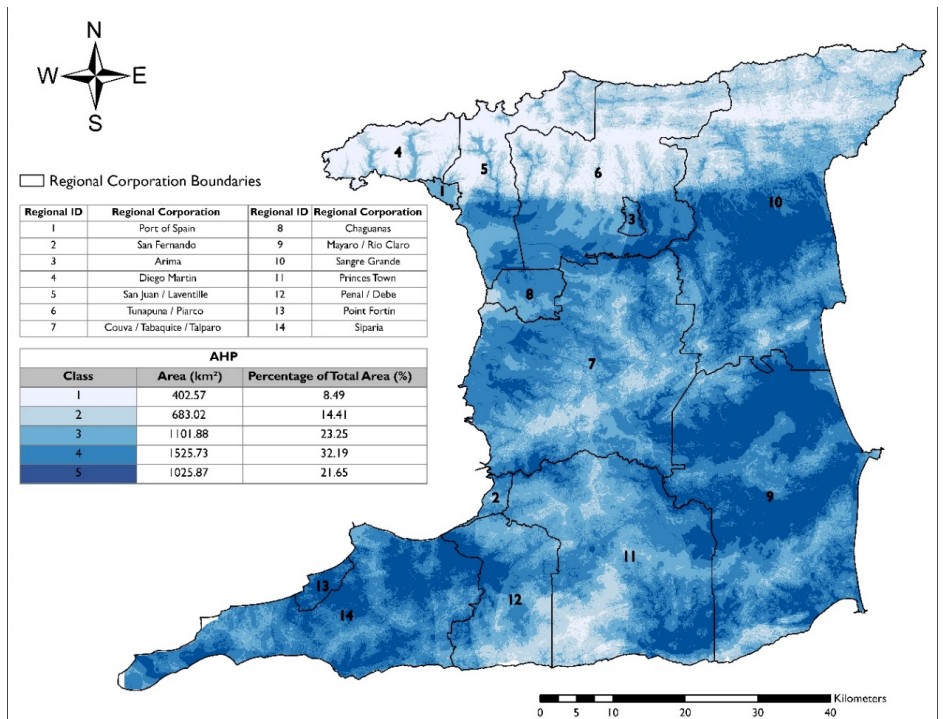

**Figure 7.** Flood Hazard Indexes created for AHP.

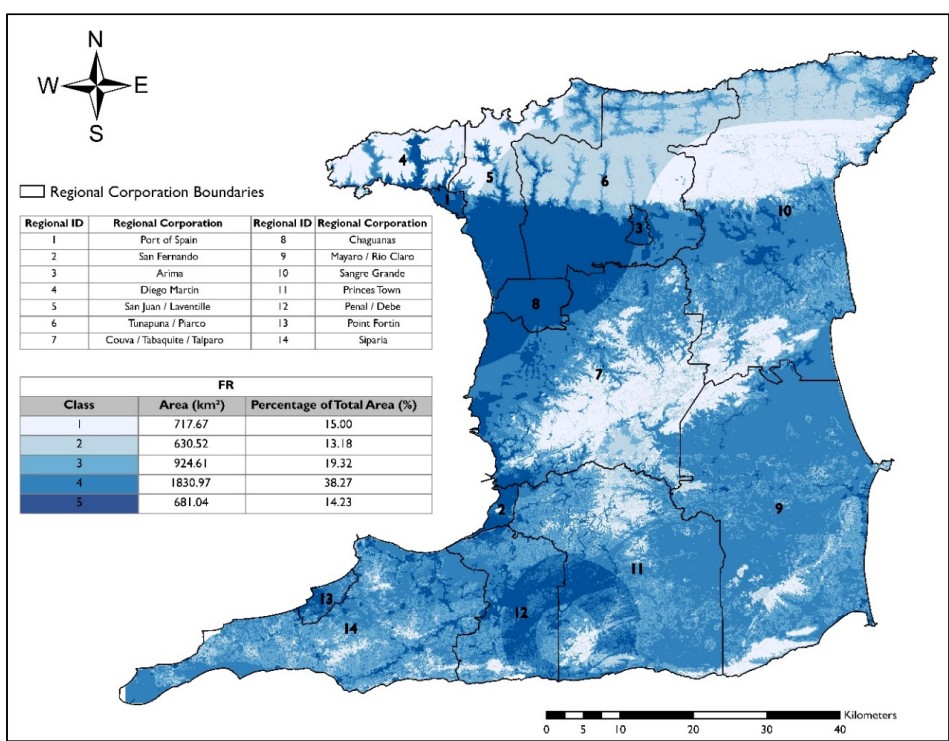

**Figure 8.** Flood Hazard Indexes created for FR.

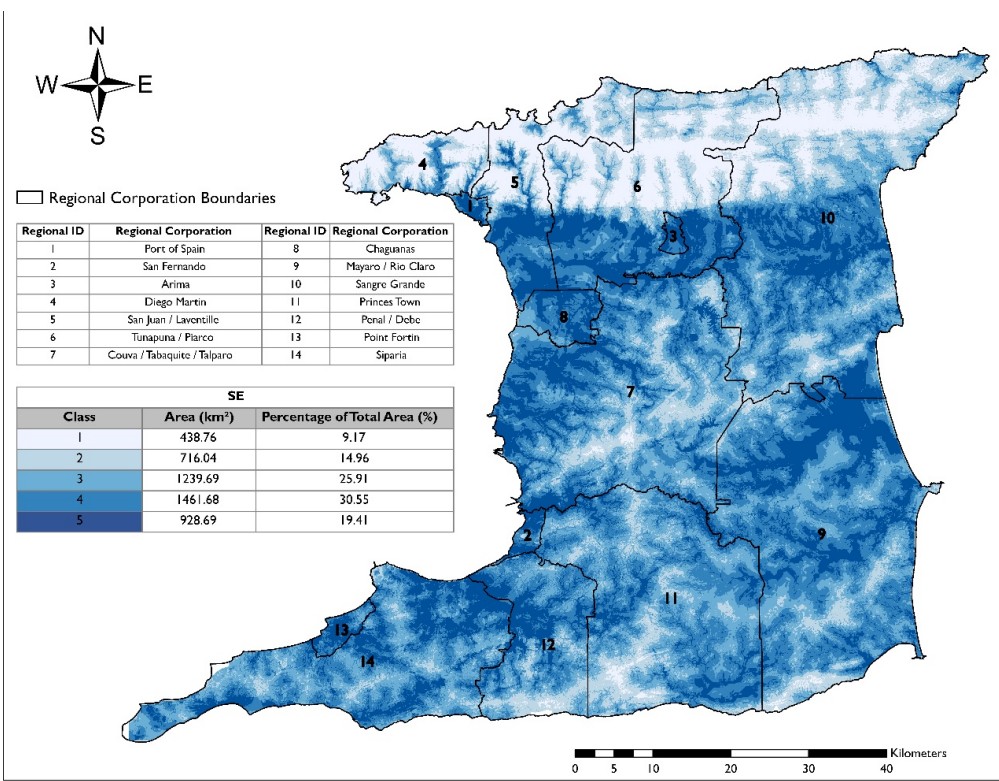

| Regional ID | Regional Corporation | Regional ID | Regional Corporation |
|---|---|---|---|
| 1 | Port of Spain | 8 | Chaguanas |
| 2 | San Fernando | 9 | Mayaro / Rio Claro |
| 3 | Arima | 10 | Sangre Grande |
| 4 | Diego Martin | 11 | Princes Town |
| 5 | San Juan / Laventille | 12 | Penal / Debe |
| 6 | Tunapuna / Piarco | 13 | Point Fortin |
| 7 | Couva / Tabaquite / Talparo | 14 | Siparia |

| SE | | |
|---|---|---|
| Class | Area (km²) | Percentage of Total Area (%) |
| 1 | 438.76 | 9.17 |
| 2 | 716.04 | 14.96 |
| 3 | 1239.69 | 25.91 |
| 4 | 1461.68 | 30.55 |
| 5 | 928.69 | 19.41 |

**Figure 9.** Flood Hazard Indexes created for SE.

### 4.1.2. FHI-FR

The slope and elevation analysis shows that there is a relationship between flooding and susceptible areas. Gentler sloping areas, that were classed with a susceptibility of 5 have an FR value higher than one (1.64) which is depicted in Table 7. It can then be inferred that lower slope values are indicative of a greater occurrence of floods. This relation can be due to the runoff of water from rivers or regions in higher slopes, collecting into the final rivers or regions in the lower slopes. Similar to slope, elevation FR values increase with increasing susceptibility classes. Elevation ranges between 481.86–923.86 metres have a value of 0, indicating that regions with this class had a minimal impact on flood occurrence. The highest and only FR value above 1 was found in class 5 (0–57.97), signifying that lower elevation has greater flood occurrences. These results signify that more flood occurrences are found in lower lying, gentler sloping regions. The FR analysis also shows a positive correlation between river density and flood occurrence. While the ratios are in close range of each other, susceptibility classes 3 and 4 display ratios >1, with the final class having a ratio near 1. This signifies that as the number of streams increases, the occurrence of flooding also increases.

Distance from rivers is a crucial determining factor of flooding. River levels increase following rainfall events, if capacity is reached during a high intensity and frequency event, overflow into surrounding areas is likely to occur. Distances between 2576–4977 m, displayed the most influence on flood occurrence. These distances can be found within the susceptibility classes 4 and 5 that have FR values of 1.02 and 1.65. This signifies that as the distance from the river increases, flood occurrence is likely to decrease.

Flow accumulation values of classes 2, 3, 4 have high-frequency ratios of 1.91, 5.44 and 3.99 respectively. This shows that there is no correlation between an increase in flow accumulation values and flood occurrences. The highest susceptibility class of a flow accumulation of 180,947.14–266,714.00, displayed FR values of 0, which signifies that this class had minimal impact on flood occurrence. This can be attributed to areas of high flow accumulation being significantly close to river systems. Since the FR analysis utilizes flood

occurrences and the frequency ratio was 0, it suggests that the occurrence data used did not fall in close proximity to rivers.

TWI is an indicator of the topographical effect on the location and size of saturated areas in generating runoff. The ratios obtained for these factors are all significantly high in comparison to the ratios of other factors. TWI values of 3.45–9.11 return an FR lower than one (0.24 and 0.72), which have a low susceptibility classing. As the TWI increases, so too the FR, signifying that there is a direct relationship between TWI and flood occurrence.

Land use determines how an area responds to an increase in runoff. Areas that are vegetated allow for interception and are less likely to flood than an area with bare land. Five land uses were identified for the susceptibility classes, where 1 is forested areas, 2 is agricultural areas, 3 is swamp areas, 4 is residential areas and 5 are water bodies. Class 4, residential, has one of the highest FR values (8.29) of all the factors. This signifies that most flood occurrences occur in residential areas. An FR value of 0 was returned for water bodies, signifying that this land use has minimal impact on flood occurrences.

CN is related to runoff and infiltration rates following excessive rainfall. Two classes, 3 and 5 have FR values of more than 1 (1.44 and 1.61), which have CN values of 72 and 100. There is no direct relationship between flood occurrence and this factor.

Lithology was a factor derived from the geology dataset. It was included in the analysis because of its influence on water behaviour on the ground. The main lithology found in the area was "poor." The highest FR (1.16) was found for the lithology with a permeability of Good and Poor. There is no direct relationship between permeability levels of the lithology and the occurrence of floods. The highest FR was found for class 2, which has good permeability. This signifies those other factors influenced the occurrence of flooding events.

The last, and most influential factor in flooding is rainfall; without rainfall, no flooding would occur. The highest FR value for all factors was found in class 2. All classes have FR more than one except class 5 (0.187–0.193), which is expected to be the most susceptible area because of high amounts of rainfall. Because the final rainfall dataset used in the analysis was an average of multiple years resulted in the FR obtained. The FR shows that there is no direct relationship between average rainfall and flood occurrence. However, if rainfall data were available for specific seasons of a particular year and the flood occurrences for that specific time, the results would reflect a correlation between the two.

The result from the FR analysis shows the influence of rainfall. Since this conditioning factor was given an FR value of 17.79 for class 2 and all other FR values were above 1, the output showed a greater influence of rainfall on the final output. This is illustrated in Figure 5b, over the Caroni and Chaguanas regions and in Penal/Debe and Princes Town regions, where the general trend of rainfall is illustrated in regions of the output map. Based on the predictive weights, derived from the frequency ratios, land use, elevation and rainfall have the highest influence on the final output. CN, lithology and river density have the lowest influence on the final output.

### 4.1.3. FHI-SE

From the SE analysis (Table 9) and output map (Figure 9), Wj results indicate that land use, distance from rivers and elevation (0.18, 0.18 and 0.15 respectively) are assigned the highest weights. River density, lithology, and CN (0.058, 0.06 and 0.07 respectively) have the lowest Wj values and are therefore assigned the lowest weights. These weights differ from AHP, where rainfall, slope and river density, which were given the highest weights in the AHP analysis, was not given the highest weights for SE. Additionally, river density was given a low SE weight similarly to FR.

SE produced results similar to AHP and this can be accounted for by the use of weights similar to AHP. The factors were not reclassified based on their integer values and as such, the general trend relating to the conditioning factors is illustrated.

From the results of all models, quantitatively, FR was determined to be the best accuracy assessment with an AUC value of 0.76. The hazard analysis identified susceptible

areas throughout Trinidad. Three areas that were highlighted as very susceptible were the regions of Chaguanas, Couva/Tabaquite/Talparo and Penal/Debe. These three areas showed greater percentages of very highly susceptible areas. Since susceptibility was determined by 10 conditioning factors that were weighted, it can be assumed that these factors influence flood occurrence in Trinidad. The number of factors used in the susceptibility analysis is important to the final output. Various studies used different numbers of factors. Ten factors were used to prevent unrepresentative weights where one specific factor dominates the influence of the model. The factors were also selected based on the environment of the entire island of Trinidad.

### 4.2. Vulnerability

After combining all vulnerability factors, three final VIs (Figures 10–12) were generated for the analysis using AHP, FR and SE. These layers were used to compute the final risk assessments. The VIs were created using the three factors that were found to majorly impact flooding in Trinidad at the island level. VIs depicted five susceptibility classes, namely, very low, low, moderate, high and very high susceptibility to flooding.

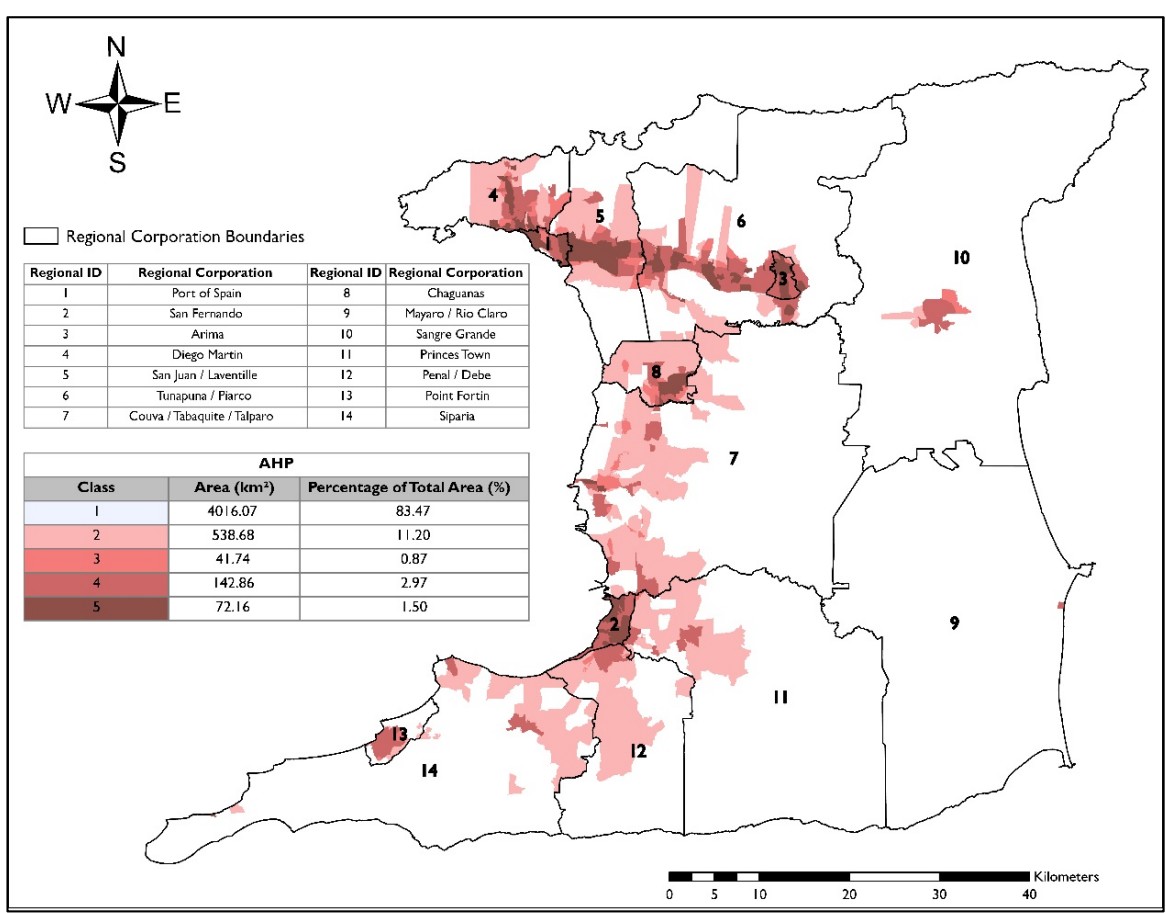

**Figure 10.** Vulnerability Indexes created for AHP.

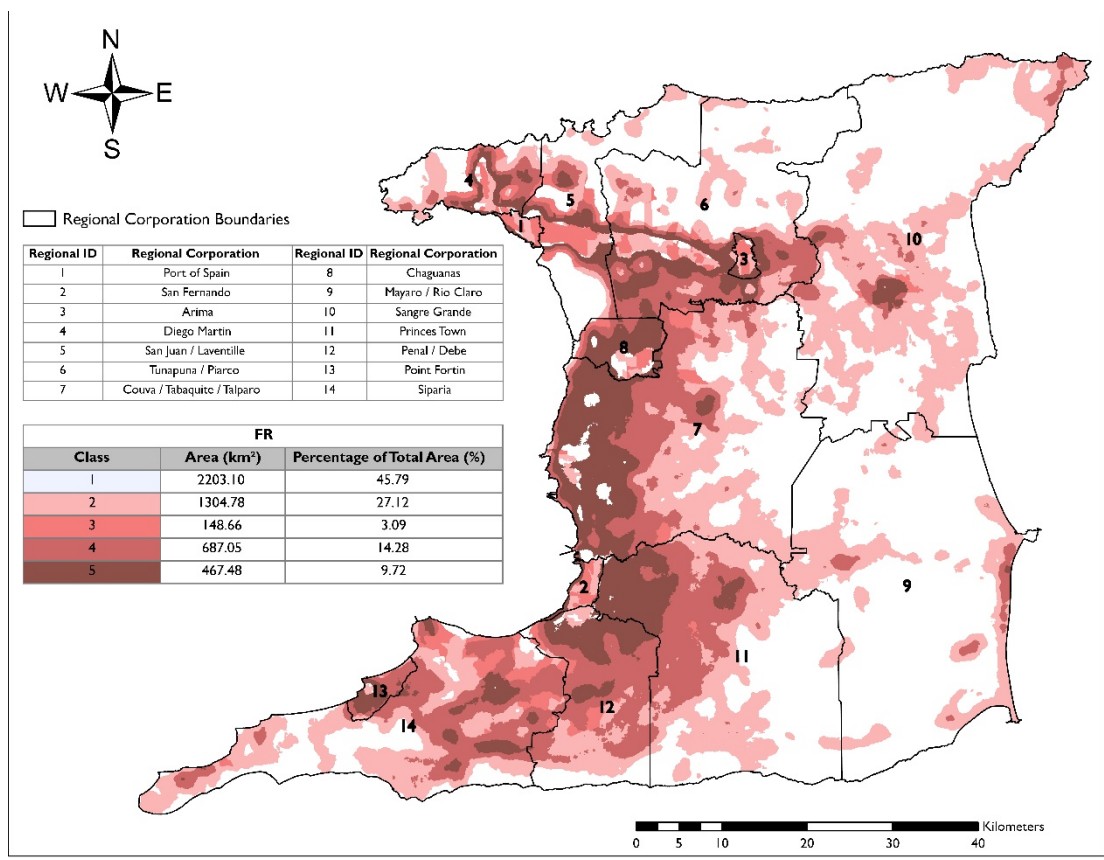

**Figure 11.** Vulnerability Indexes created for FR.

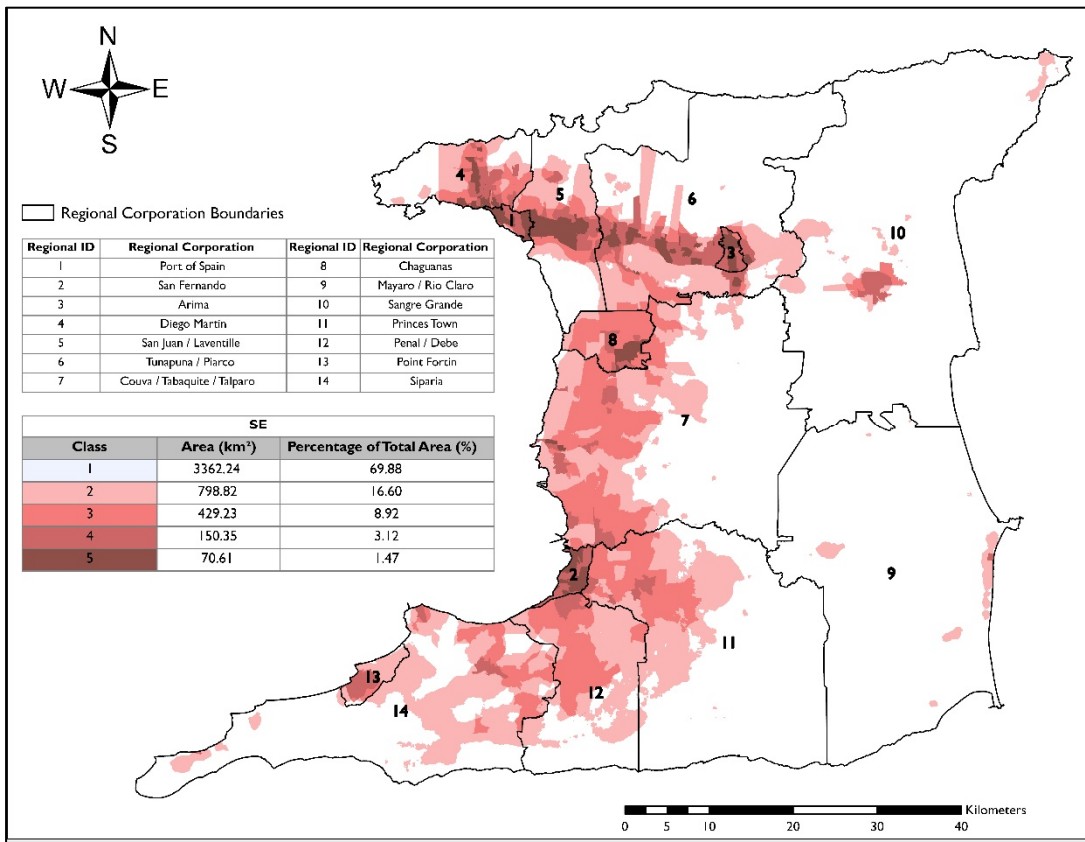

**Figure 12.** Vulnerability Indexes created for SE.

### 4.2.1. VI-AHP

The VI generated (Figure 10), illustrates high and very high vulnerability regions in Port of Spain, Chaguanas and San Fernando. The high vulnerability areas are concentrated in the western regions of Trinidad, where there is a greater population and by extension more buildings and roads. Regional Corporations such as Sangre Grande and Mayaro/Rio Claro, have lower population densities, resulting in lower building and road densities. Because of this, this region displayed very low, low and some moderate vulnerabilities to flooding.

### 4.2.2. VI-FR

Classes 2, 3 and 4 of the road density factors were found to have a greater flood occurrence since values were greater than one (1.00, 2.27 and 3.00 respectively) as shown in Table 8. This shows that areas with the highest road density and as such the greatest vulnerability class had low flood occurrences. Building density, however, had high-frequency ratios in classes 4 and 5. These values were significantly greater compared to other values (11.12 and 11.07). Population density showed a similar trend to building density, displaying values above 1 for classes 2, 3, 4 and 5. The greatest value, 10.39 was found in class 5, signifying that class 5 had the most flood occurrences compared to the other classes of this factor. Frequency Ratio displays a similar trend to AHP, with more vulnerable areas being identified in the western parts of Trinidad. However, a greater percentage of areas (14.28 and 9.72%) are identified as high and very-high susceptibility to flooding (Figure 11). There is also a greater percentage of areas identified as high vulnerability compared to very-high vulnerability.

### 4.2.3. VI-SE

Table 8 shows that similarly to AHP, SE gave a greater weight (0.50) to population density than FR, however, road density and building density did not have the same trend. The lowest weight (0.24) was given to building density, however, the lowest weight for AHP was given to road density (0.16). Similarly, to AHP, SE identifies regions in the East–West corridor, Chaguanas and San Fernando with very-high susceptibility (Figure 12). It also follows a trend similar to both AHP and FR. The results also depict that there is a greater percentage of areas that have a high vulnerability (3.12%) compared to very-high vulnerability (1.47%), like FR.

### 4.3. Risk Assessment

Figures 13–15 illustrate the risk maps produced from the susceptibility and vulnerability analysis. It also identifies the various risk classes within the Regional Corporations of Trinidad, the contributing percentages to the overall risk for the entire island and the individual risk classes for each of the Regional Corporations. The watersheds that contain very-high risk areas are also identified. The risk maps were created by combining the FHIs and Vis using Equation (12):

$$Risk = Hazard \times Vulnerability \tag{12}$$

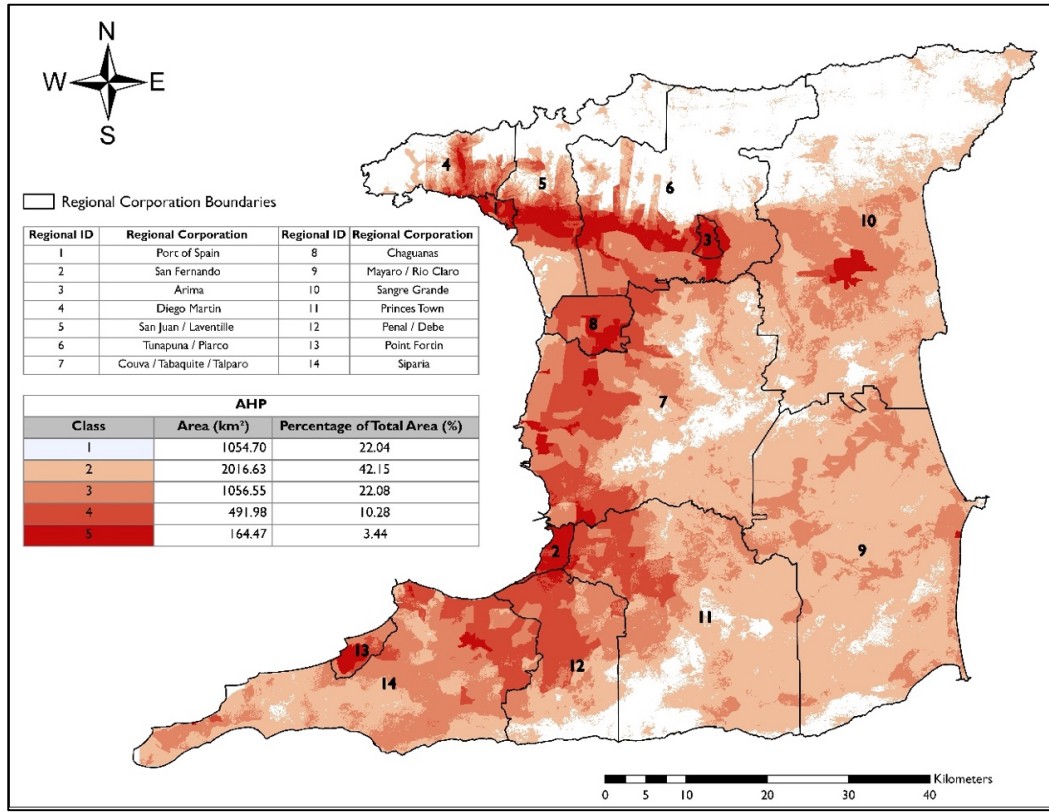

**Figure 13.** Risk assessments created for AHP.

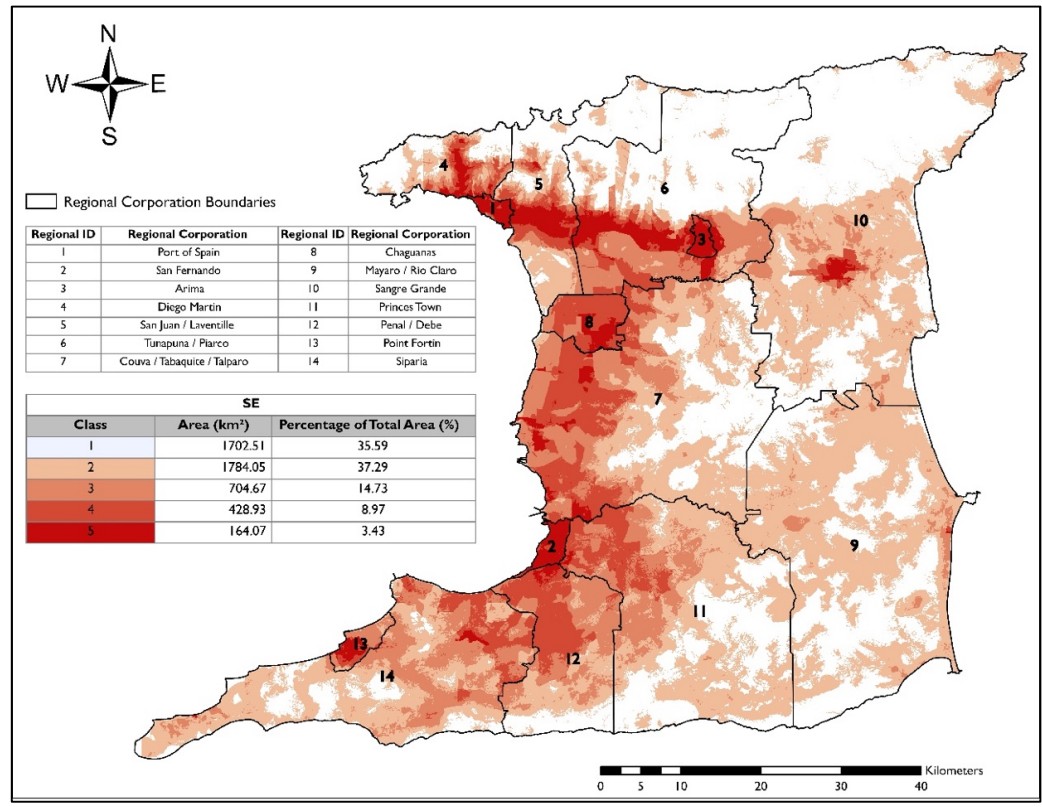

**Figure 14.** Risk assessments created for FR.

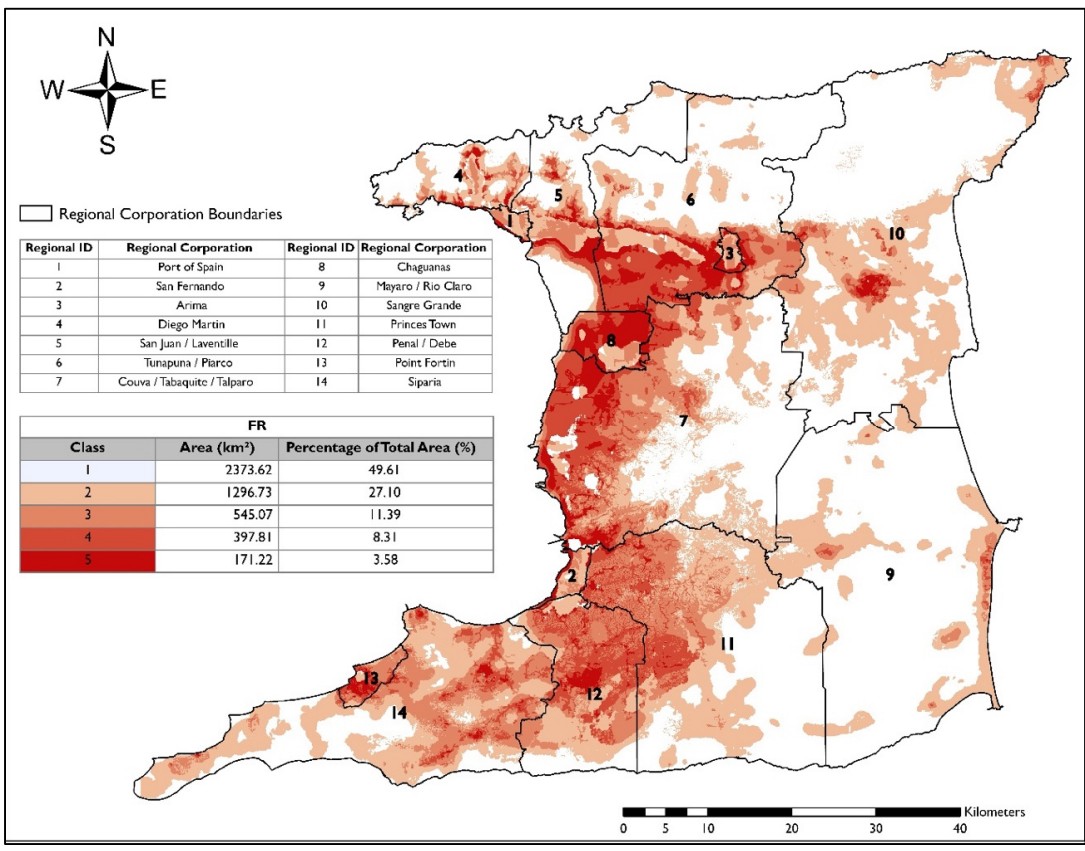

**Figure 15.** Risk assessments created for SE.

*4.4. Validation*

Area Under the Curve (AUC) was used to validate the susceptibility and risk results obtained for each weighting technique, in which FR depicted the best AUC for both. The AUC values obtained for the validation of risk were lower than susceptibility due to the incorporation of two components, hazard and vulnerability in the final output. Since AUC generates a value to assess the appropriateness of the results in relation to occurrence data, and the vulnerability layer was not prepared with flood occurrence data, the success rates for the final risk outputs were lower compared to that of susceptibility outputs.

The success rate of the various models was found to vary, FR and SE produced the best results compared to AHP. This is also shown by the steepness of the various ROC curves depicted in Figures 16–18. According to the results of the ROC curves, the FR, which had the greatest steepness of all curves was found to have an AUC classification of 0.64 (Figure 17), which although considered satisfactory is still low and far from the perfect classification of 1. AHP and SE model outputs produced ROC curves that were gentler with AUC values of 0.51 and 0.57 respectively. An AUC of 0.5 classifications is considered to have results that reflect less association to actual occurrence, signifying it occurred by chance. Therefore, the AHP and SE results displayed satisfactory accuracy.

Albeit the method proves to be effective for determining the best suitable weighted model for risk assessment and its components by the good agreement between susceptible zones and flood occurrence points, limitations and uncertainties for future work should be discussed. These limitations are related to the data scarcity that exists in Trinidad, similarly to most SIDS, which can be a challenge for future studies.

The first limitation of the study is the lack of rainfall data for the study area. There is only one station that collects rainfall data which, if used, would have been inadequate in representing the meteorological spatial variation at the scale of the study. As such, TRMM rainfall data was utilized which has a spatial resolution of 30 m and no calibration of the data was conducted because of the lack of studies that utilized these data. Addressed by

Harris et al. [39], is the need for a more rational and regime-based adjustment procedure of satellite data to increase its reliability.

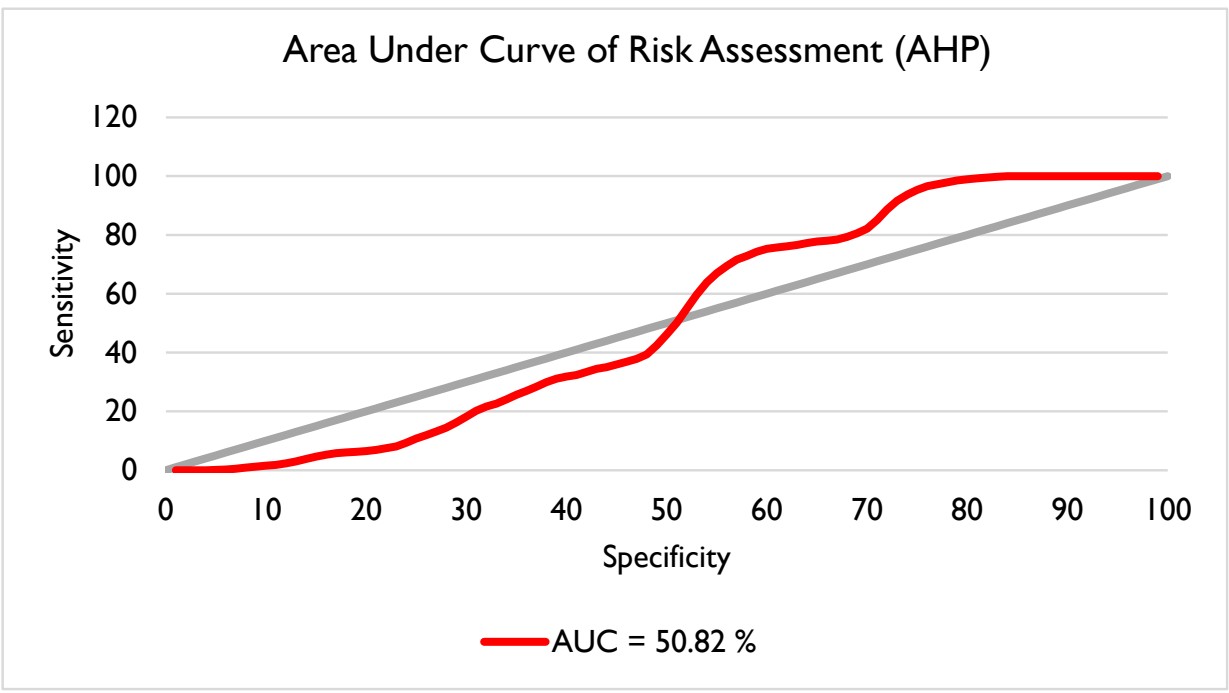

**Figure 16.** Area Under the Curve conducted for risk assessment created from AHP.

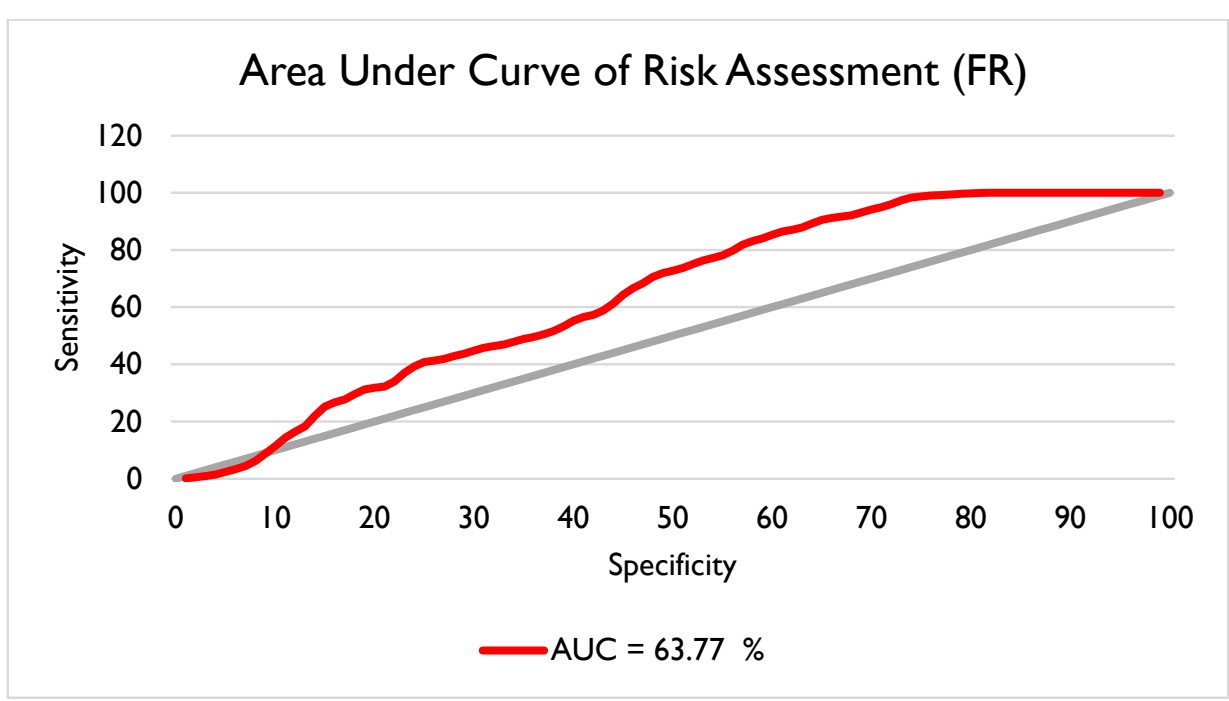

**Figure 17.** Area Under the Curve conducted for risk assessment created from FR.

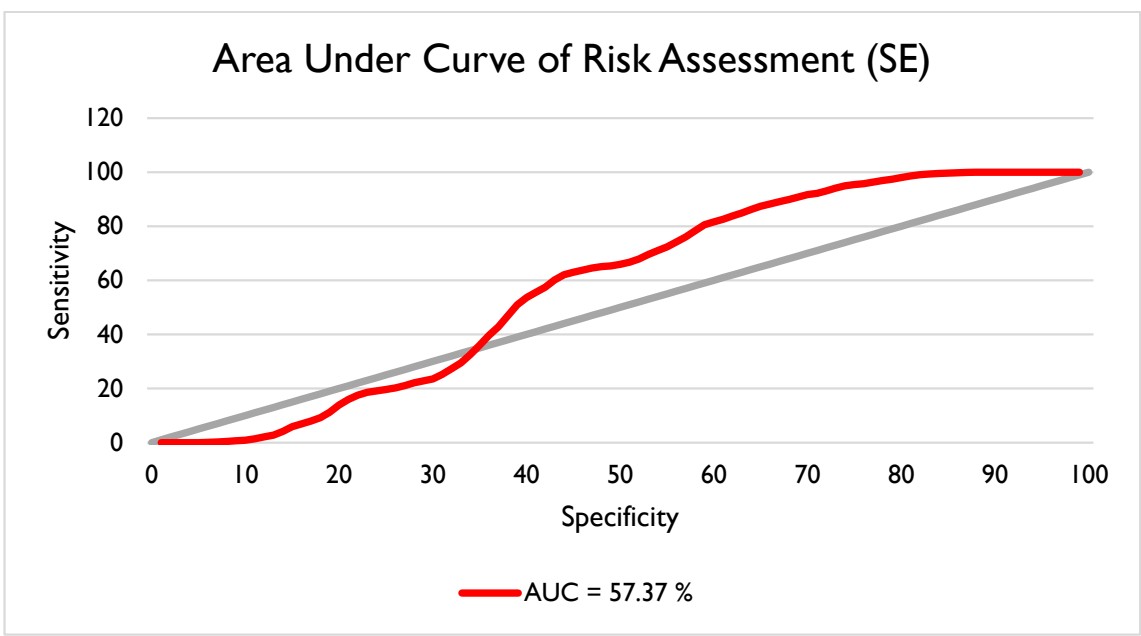

**Figure 18.** Area Under the Curve conducted for risk assessment created from SE.

Another important issue is the lack of flood data occurrence. There is no national spatial data infrastructure to combine all occurrences of flooding in Trinidad. This is an important aspect, that contributes decisively to more realistic flood risk assessments [40]. While there was a diverse amount of occurrence data, ranging from 2000–2019, all records of flooding in Trinidad were not encompassed. Additionally, some of the occurrence data collected were unconfirmed reports from residents of Trinidad.

Furthermore, is a lack of census data to be used for vulnerability assessments. An outdated population dataset (2011) was used and as a result of the lack of census data, more factors for the vulnerability were not assessed, which should include criteria such as elderly; children four years and younger; single parents; low-income people, persons with tertiary level education; and rented apartments (Rincón, Khan and Armenakis 2018). These vulnerability criteria should incorporate physical, environmental, social and economic components.

*4.5. Conclusions*

The study aimed at assessing flood risk in Trinidad using multiple weighting techniques, namely, AHP, FR and SE to create flood hazard and vulnerability indexes. 10 conditioning factors that encapsulate the diverse and unique physical environment of Trinidad were used. The results were validated using occurrence data to ascertain which had the best accuracy. Risk results indicate that FR had the highest accuracy (64%) followed by SE (57%) and AHP (51%). Susceptibility assessment indicated that FR also had the highest accuracy (76%), showing that there is good agreement between the results and the flood occurrence data. The findings were consistent with the findings of previous research and emphasizes the need to quantitatively assess flooding potential in Trinidad. Studies related to risk management have been conducted in the past however, existing research does not provide enough of a robust method for assessing flood at a microscale specific to the unique environment of SIDS. By identifying the most suitable weighting technique, this study provides a crucial starting point for flood risk management and decision-making by state agencies, where high-risk areas can be identified and prioritized for further quantitative assessments at a micro scale. The study may be used to inform risk identification, risk reduction, preparedness, land use planning, financial protection and resilient reconstruction. The results use the communities in Trinidad as a case study, to reflect other similar communities in SIDS, thereby providing a roadmap for what is required at a community level. In most developing and tropical nations, flood baseline information is usually limited. This leads to

the need for practical methodologies to be applied for appropriate and repeatable disaster risk assessments. This is applicable to SIDS where the vulnerabilities to increased effects of climate change are high. Notwithstanding the limitations of the study, the method can be modeled in areas that have the same physiography, hydrometeorology, geomorphology, size and vulnerabilities as Trinidad. This study seeks to sufficiently represent the geostatistical and cumulative nature of the parameters that influence flooding in Trinidad as it relates to the limitations and inherent characteristics of SIDS and developing countries. It is also adding to the research currently conducted in the Caribbean by providing different modelling strategies and approaches that will contribute to the body of studies in the future.

**Author Contributions:** Conceptualization, C.R. and B.R.; methodology, C.R.; software, C.R.; validation, C.R., B.R. and R.R.; formal analysis, C.R., B.R. and R.R.; investigation, C.R.; resources, C.R. and B.R.; data curation, C.R. and B.R.; writing-original draft preparation, C.R.; writing-review and editing, C.R., B.R. and R.R.; visualization, C.R.; Supervision, B.R. All authors have read and agreed to the published version of the manuscript.

**Funding:** This research received no externation funding.

**Data Availability Statement:** The data that support the findings of this study are available from the corresponding author, [R.R.], upon reasonable request.

**Conflicts of Interest:** The authors declare no conflict of interest.

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
