# Peer review of "A Comparative Analysis of Weighting Methods in Geospatial Flood Risk Assessment: A Trinidad Case Study"

_land, doi:10.3390/land11101649_

Round 1
Reviewer 1 Report
Please see attached document

Author Response
Thank you for your comments and suggestions. Please see attached document.

Reviewer 2 Report
The article is clear in its scientific reading and interpretation. The qualitative study is presented in interesting approach linked to comparative analysis of weighting methods in geospatial flood risk assessment.
The conceptual discussion about flood vulnerability and risk assessments should be more in-depth and with more bibliographic references making the comparison with another Studies in islands of the same latitudes.
Some figures will need to be improved in terms of visual and graphic quality (figures 3, 4, 5, 6 and 7).
The conclusions should be improved in order to present advantages of the analysis on Flood hazard and vulnerability indexes presented in the face of climate change. Include, also, the difficulties and disadvantages of the methodologies and techniques used, as well as the challenges and lines in future research.
Author Response
Thank you for your comments and suggestions. Please see attached document

Reviewer 3 Report
Review of manuscript "A comparative analysis of weighting methods in geospatial flood risk assessment: A Trinidad case study" (land-1892887)
Dear authors, your research aims to compare three different weighting methods in geospatial flood hazard risk assessment. It is a well-prepared manuscript and fits the aims and scope of the journal topic. Nevertheless, the authors need to highlight the novelty of their research as compared with previous research. Therefore, "Major Revision" is required to improve this manuscript. Specifically, the reviewer has the following comments and suggestions:
(1) The Introduction Section: this part is very weak because the authors did not highlight the purpose and novelty of this study from an international perspective. As a consequence, reviewers cannot figure out why this research must be performed in this context. If this research just presented a case study in Trinidad and Tobago, then it lacks novelty for publishing in this internationally-distinguished journal. I would like to remind that these three weighting methods are not new in flood risk assessment.
(2) The Literature Section: in this part, the authors must look further into the latest research about flood risk assessment and waterlogging risk assessment. In particular, the advanced maximum entropy method has been largely used in flood risk assessment (please find below). Unfortunately, this new method was totally ignored in this manuscript. A thorough literature review is meant to set the context for your research work and highlight how it contributes to the knowledge in this field and builds on previous relevant research.
https://doi.org/10.1016/j.scs.2022.103812
https://doi.org/10.1080/10106049.2017.1316780
https://doi.org/10.1007/s11069-020-04453-3
(3) Figure 2: this figure could be largely improved by using different colors.
(4) In Methodology Section: in this part, the authors did not present all the details of the input data, such as the dates in acquiring them, accuracies, and spatial resolution. The authors should largely improve their statements and expressions.
(5) I also suggest the authors to provide a flowchart of the methodology part.
(6) The authors should also collect and build a long-term flood risk inventory, which could increase the reliability of the risk assessment results.
(7) Figure 3: the scales of these maps could be integrated into a single scale.
(8) Section 3.2, Classification: in this part, most of the contents belong to the literature review of conditioning factors, which should be moved to the Literature Section.
(9) Tables should be "three-line table" without any color.
(10) Table 5: how to determine the values in different classes?
(11) Figure 5: I suggest the authors to overlay these three different maps so that the differences between them can be revealed more clearly.
(12) Section 4.4, Validation: the Area Under the Curve (AUC) of this research is very low indeed. Please double check the results. Generally speaking, previous research shows that only an AUC over 0.80 can be deemed “outstanding”.
(13) All the contents in the template should be removed from the submitted manuscript.
Author Response

(The authors gave the same response as above.)

Round 2
Reviewer 1 Report
Please see attached file

Reviewer 2 Report
The authors have made an effort to adequately address the suggestions made as a reviewer.
Reviewer 3 Report
The manuscript is acceptable after the English language has been largely improved.
